

SciPost Phys. 1(1), 004 (2016)

# Quantum-optical magnets with competing short- and long-range interactions: Rydberg-dressed spin lattice in an optical cavity

**J. Gelhausen, M. Buchhold, A. Rosch and P. Strack**[*],

Institut für Theoretische Physik, Universität zu Köln, D-50937 Cologne, Germany

* pstrack13@gmail.com, https://arxiv.org/a/strack_p_1.html

## Abstract

The fields of quantum simulation with cold atoms [1] and quantum optics [2] are currently being merged. In a set of recent pathbreaking experiments with atoms in optical cavities [3, 4], lattice quantum many-body systems with both, a short-range interaction and a strong interaction potential of infinite range –mediated by a quantized optical light field– were realized. A theoretical modelling of these systems faces considerable complexity at the interface of: (i) spontaneous symmetry-breaking and emergent phases of interacting many-body systems with a large number of atoms $N \to \infty$, (ii) quantum optics and the dynamics of fluctuating light fields, and (iii) non-equilibrium physics of driven, open quantum systems. Here we propose what is possibly the simplest, quantum-optical magnet with competing short- and long-range interactions, in which all three elements can be analyzed comprehensively: a Rydberg-dressed spin lattice [5] coherently coupled to a single photon mode. Solving a set of coupled even-odd sublattice master equations for atomic spin and photon mean-field amplitudes, we find three key results. (R1): Superradiance and a coherent photon field appears in combination with spontaneously broken magnetic translation symmetry. The latter is induced by the short-range nearest-neighbor interaction from weakly admixed Rydberg levels. (R2): This broken even-odd sublattice symmetry leaves its imprint in the light via a novel peak in the cavity spectrum beyond the conventional polariton modes. (R3): The combined effect of atomic spontaneous emission, drive, and interactions can lead to phases with anomalous photon number oscillations. Extensions of our work include nano-photonic crystals coupled to interacting atoms and multi-mode photon dynamics in Rydberg systems.

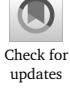
# 1  Introduction and key results

The fields of quantum simulation with cold atoms [1] and quantum optics [2] are currently being merged. On the one hand, to coherently couple photons to low-entropy, correlated quantum many-body states –the objects of desire of quantum simulators– offers new possibilities to imprint atomic coherences and quantum correlations onto quantum light such as dissipatively anti-bunched photons [6] or a "many-fermion" EIT [1] window without light absorption [7]. On the other hand, the inter-atomic interactions mediated by the photons opens up explorations of previously inaccessible phases such as long-ranged quantum spin- and charge glasses [8–12], bond-ordered phases [13], dynamical spin-orbit couplings [14, 15], or topological states carrying perpetual currents with dynamical gauge couplings [16, 17].

    A major objective in the field of quantum simulation is to prepare and probe low-entropy quantum magnets. Existing efforts have focussed on magnetic interactions via superexchange [18], mapping to charge degrees of freedom [19], dipolar interactions [20] with polar molecules [21], magnetic atoms [22], and laser-dressed Rydberg atoms [5]. More complex interaction potentials necessary for frustrated magnetism have also been proposed [23–25]; optical pumping schemes should allow to access non-equilibrium magnets, too [26, 27]. One common goal of these efforts is the engineering of a *single magnetic interaction rate with a certain angle-dependence and range*, which can compete with kinetic energies, longitudinal fields, and the

---

[1]Electromagnetically Induced Transparency

decay processes.

In this article, we want to initiate the study of *quantum-optical magnets with competing short- and long-range interactions*, the latter being mediated by a dynamical photon field. One may wonder how a single spin or quantum dipole (with in principle fixed charge distribution) can interact via *two competing potentials with drastically different range* and independently tunable magnitude: At the core of our proposal is an atomic "two dipoles in-one" unit (illustrated below in Fig. 7), to which –depending on the principal quantum number and electronic transitions used– two different force-mediating photon fields can couple simultaneously (described in Sec. 2).

In addition to the novel magnetic phases an array of such "two dipoles in-one" can attain, a key question is how the quantum dynamics of the photon field is affected by the magnetic correlations. Using an optical cavity for one of the photonic force carriers and a deep optical lattice to freeze out the motion of the atoms is a natural experimental set-up, which will allow non-destructive detection of purely magnetic correlations via the cavity output spectrum [3, 4, 28]. Let us note that the question of how quantum light interacts with a self-interacting set of qubits is of broader relevance including for example cavity Rydberg polaritons [29, 30], Rydberg-EIT setups [6,31,32] and nano-photonic devices [33–35]. Especially in reduced dimensions with confined electric fields, even small qubit-qubit interactions can have a huge effect.

These systems generate a lot of complexity at the interface of three typically only weakly connected areas of physics: (i) *emergent phases and spontaneous symmetry-breaking* of interacting many-body systems in the thermodynamic limit ($N \to \infty$ number of qubits) (ii) *quantum optics and the dynamics of fluctuating light fields* (from $M = 1$ to $M = \infty$ photon modes), and (iii) *non-equilibrium physics of driven, open quantum systems*, due to drive and multiple loss channels such as photon decay with rate $\kappa$ and atomic spontaneous emission with rate $\gamma$.

The goal of the present paper is to provide a "base case" or the simplest prototype of a quantum-optical magnet with competing short- and long-range interactions, in which the interplay of the above mentioned (i)-(iii) can be transparently studied.

## 1.1 Model: Rydberg-dressed spin lattice coupled to single-mode optical light field

As a suitable model (Fig. 1), we propose to supplement the existing experimental set-ups [3,4] by weakly admixing a Rydberg-level with relatively low principal quantum number ($n \sim 30$). The other, but equivalent, point of view is to couple a Rydberg-dressed spin lattice [5] to a single mode of an optical resonator. See also Refs. [36] for a related setup but without a lattice. As we derive below in Sec. 2, the pure spin-part of the Hamiltonian $H = H_{\mathrm{spin}} + H_{\mathrm{spin-light}}$ is

$$H_{\mathrm{spin}} = -\frac{\Delta}{2} \sum_{\ell=1}^{N} \sigma_{\ell}^{z} + \frac{V}{d} \sum_{\langle \ell m \rangle} \left( \frac{1 + \sigma^{z}}{2} \right)_{\ell} \left( \frac{1 + \sigma^{z}}{2} \right)_{m} , \tag{1}$$

where the sum $\sum_{\langle \ell m \rangle}$ goes over all nearest-neighbor pairs of the square lattice and $d = z/2$ is the dimension of the lattice with $z$ the coordination number. For a negative ($-\Delta < 0$) longitudinal field it is favorable for the spins to point up along the $z$-axis $| \uparrow \rangle$. Competing against this is the repulsive or antiferromagnetic "Rydberg-mediated" term, which minimizes energy by pushing the spin in spatially alternating configurations, e. g. $| \uparrow\downarrow\uparrow\downarrow ... \rangle$. In contrast to a conventional Ising $\sim \sigma_{\ell}^{z} \sigma_{m}^{z}$ interaction term, the Rydberg interaction is conditioned on population in the upper state.

Non-trivial quantum fluctuations are added to $H_{\mathrm{spin}}$ by coherent conversion of spin excita-

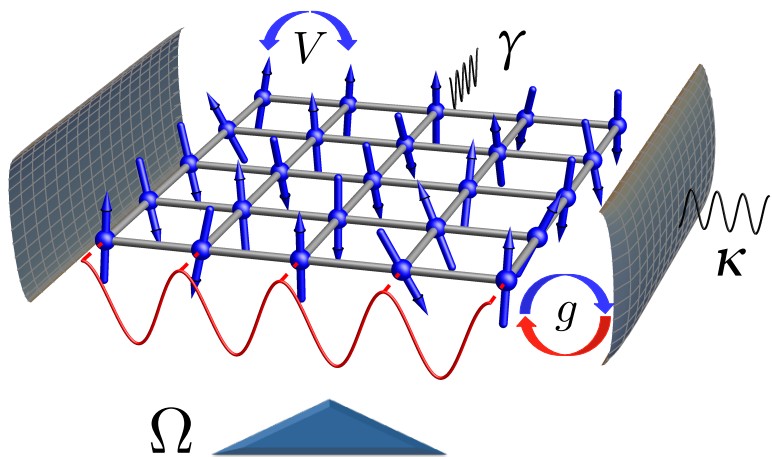

Figure 1: Rydberg-dressed spin lattice coupled to single-mode optical light field. We take the atoms to be in a deep optical lattice (for example a Mott insulator at unit filling) such that no motion occurs and only internal spin excitations drive the dynamics. From weakly admixing a Rydberg level, an effective nearest-neighbor interaction (repulsive) $V$ competes with an effectively infinite range interaction from the single light mode of the cavity that couples all atoms with strength $g$. The atoms can spontaneously decay with rate $\gamma$ and photons can leave the system through the cavity mirrors with rate $\kappa$. The system is driven from the side with a laser of Rabi frequency $\Omega$ to deposit excitations into the system. Driven dipoles in other lattice geometries are also interesting to consider [37].

tions into photons with rate $g$

$$H_{\text{spin−light}} = \frac{g}{\sqrt{N}}(a + a^{\dagger}) \sum_{\ell=1}^{N} (\sigma_{\ell}^{+} + \sigma_{\ell}^{-}) + \omega_0 a^{\dagger} a \,, \tag{2}$$

where $\omega_0$ is the effective cavity frequency in a rotating frame and $N$ is the number of atoms; the rescaling of the effective spin-light coupling with $1/\sqrt{N}$ ensures a non-trivial thermodynamic limit (i.e. taking the system size and number of atoms to infinity keeping the atomic density and electric field strength per volume constant). All coupling constants appearing here are expressed in terms of fundamental quantum-optical parameters in Sec. 2 and $g$ is proportional to the external laser drive $\Omega$. Note that the $(a + a^{\dagger})$ may be viewed to act on the spins like a "transverse field" in $x$-direction, whose value depends on the quantum state of the photons. Quantum optically, both, the co- and counter rotating terms appearing in Eq. (2) are naturally induced by the cavity-assisted Raman transitions, see also Sec. 2. Within our mean-field treatment, the light-field in the cavity can either be the vacuum mode or it can be in a coherent state. By cavity vacuum, we mean the Fock state with zero photon excitations: $a|0\rangle = 0$. Then, $\langle a + a^{\dagger} \rangle = 0$ and the spins see zero transverse field. If there is macroscopic occupation of the cavity mode, then $\langle a + a^{\dagger} \rangle \neq 0$ and the system is in a superradiant state.

Our model is completed by the inclusion of Lindblad operators for photon losses through the mirrors with rate $\kappa$ and spontaneous emission of the atoms with rate $\gamma$ into the reservoir modes of the electromagnetic vacuum surrounding the cavity:

$$\mathscr{L}_{\gamma}[\rho] = \frac{\gamma}{2} \sum_{\ell=1}^{N} \left[ 2\sigma_{\ell}^{-} \rho \sigma_{\ell}^{+} - \{\sigma_{\ell}^{+}\sigma_{\ell}^{-}, \rho\} \right], \tag{3}$$

$$\mathscr{L}_{\kappa}[\rho] = \kappa \left[ 2a\rho a^{\dagger} - \{a^{\dagger}a, \rho\} \right], \tag{4}$$

| Phase | Broken Symmetry | Order Parameter |
|---|---|---|
| $\text{SR}_{\text{UNI}}$ | Superradiance $\mathbb{Z}_2$ | $\langle a \rangle \neq 0$ |
| AFM | Lattice translations $T_{\text{lat}}$ | $\langle \sigma_e^z \rangle - \langle \sigma_0^z \rangle \neq 0$ |
| AFM+SR | $\mathbb{Z}_2$ and $T_{\text{lat}}$ | $\langle \sigma_e^z \rangle - \langle \sigma_0^z \rangle \neq 0$, $\langle a \rangle \neq 0$ |
| FP | None | None |

Table 1: Order parameters for the phases in Fig. 2. Whenever the photon parity is broken, the $x$-projections of the spins also attain a finite expectation value $\langle \sigma^x \rangle \neq 0$.

where $\rho$ is the system density matrix. Spatially modulated phases in the presence of coherent driving of lattice atoms have been discussed in an open, non-equilibrium setting in particular by Lee and collaborators [26,27]; see also Ref. [38]. Here, we extend such models by coupling the spin degrees of freedom to a quantum light field, which can also be in a zero-photon vacuum state with undetermined phase. In fact, the loss rate for the photons $\kappa$ wants to drive the photons into this vacuum state (i.e. the empty cavity $|0\rangle$).

Quite generally, driven dissipative lattice models are currently under intense investigation. The systems range from (effective) spin-1/2 [26,27,39,40] and Bose-Hubbard [41,42] models to systems with interacting photons in cavity arrays [43–46].

We now present our main results from an analysis of Eqs. (1-4) using even-odd sublattice mean-field master equations (derived in Sec. 3) and the input-output formalism. A detailed discussion and derivation of these results can be found in Sec. 4.

## 1.2 Result 1: Combination of superradiance and magnetic translation symmetry-breaking

Our first key result is Fig. 2: the non-equilibrium steady-stase phase diagram of Eqs. (1-4) setting the atomic spontaneous emission $\gamma$ to zero for now. Using cavity-assisted Raman transitions [47] to tune the atom-light coupling, this describes the limit of relatively far detuned excited states, where population in the decaying levels is suppressed. These phase diagrams are computed from solving for steady states of mean-field master equations for the real-valued atomic variables ($\langle \sigma^x \rangle, \langle \sigma^y \rangle, \langle \sigma^z \rangle$) and the complex-valued photon expectation values ($\langle a \rangle, \langle a^\dagger \rangle$), see Eqs. (12-15).

This way of solving the problem implicitly takes first the thermodynamic limit $N \to \infty$ and subsequently the long-time limit $t \to \infty$. We keep $\kappa$ finite to account for photon losses. In App. A, we show how the somewhat unphysical limit $\kappa \to 0$ reproduces in fact the phase boundaries of a corresponding ground state $T = 0$ model. In a quantum optics experiment, this most closely seems to correspond to the preparation protocol in which the interacting spins are prepared in a low-entropy state in a given phase, first without any coupling to the cavity (i.e. the transversal laser drive $\Omega$ turned off). Starting in the AFM phase in Fig. 2, for example, the coupling $g$ is then turned on to induce superradiance and the AFM+SR phase. However, existing experiments and in particular the onset of superradiance are surprisingly robust against variations in the preparation scheme [3,4,51].

The phases shown in Fig. 2 can be classified according to their "order parameters" in Table 1. Let us describe the phases in more detail. Upon increasing the coupling to the photons along the $g$-axis in Fig. 2, for $V/|\Delta| < 1/4$, a fully polarized phase (FP$_\uparrow$, $|\uparrow\uparrow\dots\rangle$) becomes superradiant crossing the Dicke transition, which has been studied in detail for both, the closed thermal and ground states as well as the open variant (most recently including also single-site atomic spontaneous emission [52,53]). The symmetry, which is spontaneously broken is

$$\mathbb{Z}_2 : [a + a^\dagger, \sigma_\ell^x, \sigma_\ell^y] \to [-(a + a^\dagger), -\sigma_\ell^x, -\sigma_\ell^y], \tag{5}$$

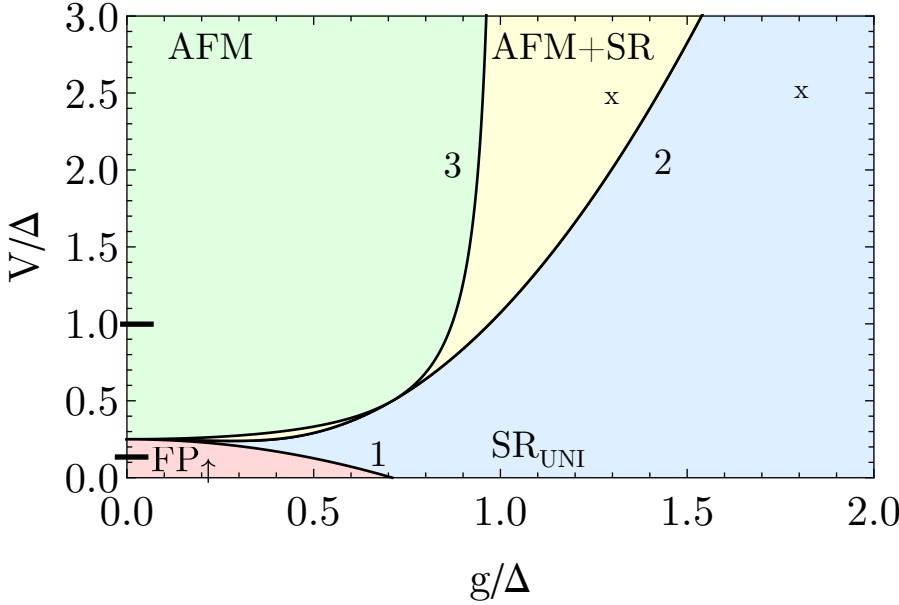

Figure 2: Non-equilibrium mean-field phase diagram of a Rydberg-dressed spin lattice (with nearest neighbor interaction $V$) coupled to a single-mode optical light field with rate $g$. In units of the spin longitudinal field $\Delta$. The key feature is the yellow strip, AFM+SR phase, in which spatially modulated magnetic moments occur together with a superradiant photon condensate, see Tab. 1. This phase may be regarded as the magnetic analogue of the (super-radiant) supersolid of moving lattice bosons in an optical cavity [3, 48–50]. The yellow strip merges into the $V$-axis at a multi-critical point from which four different phases can be reached by infinitesimal variation of parameters. At the multi-critical point ($g/\Delta = 0$, $V/\Delta = 1/4$) the spins are "maximally soft", i. e., they feel zero effective, longitudinal field. Here this in-finitesimal sensitivity to ordering is not rounded off by cavity decay induced noise. Recall that typically the cavity-induced decay rate for an atom is $\sim g^2\kappa/(\omega_0^2 + \kappa^2)$. Here however the noise kicks of the two photon absorption and emission pathways destructively interfere in the level scheme Fig. 7, which, in turn, manifests itself in the $\sim g(\sigma_\ell^+ + \sigma_\ell^-)(a + a^\dagger)$ coupling. $SR_{UNI}$ is a uniform superradiant phase in which the spins also develop an expectation value in $x$-direction. AFM stands for antiferromagnetic with differing magnetic moments on the even and the odd sublattice. $FP_\uparrow$ is a fully polarized phase in which all spins point up. The magnetisations and the value of the photon condensate across the transitions are continuous. Cavity spectra at positions labeled with (x) are depicted in Fig. 9. Numerical parameters used: $\omega_0/\Delta = 2.0$, $\kappa/\Delta = 0.2$.

The experimental signature is a jump of the photon number inside the resonator [3,52]. Recall that here we have the atomic spins pinned in an optical lattice, which takes away the photon recoil momenta transferred from the photons to the atoms. This is in contrast to the realizations of the Dicke model with momentum states of the atomic gas [51,54,55]; therein the onset of superradiance is accompanied by even-odd checkerboard formation. Here, with the setup Fig. 1, the superradiance leads to uniform spin polarization in $x$-direction and the Rydberg-mediated repulsion *competes* with this and tends to break the even-odd lattice translation symmetry.

Up the $V$-axis, at $g = 0$ the magnetisations of the system can change discontinuously (at $V = \Delta/4$ in two dimensions) from a fully polarised state (FP$_\uparrow$) to an antiferromagnetic excitation pattern (AFM). This AFM phase breaks a discrete even-odd translation symmetry

$$T_{\text{lat}} : [\sigma^\alpha_{e,o}] \to [\sigma^\alpha_{o,e}] , \tag{6}$$

which can lead to different sublattice magnetisations as depicted in Fig. 2. Here, $T_{\text{lat}}$ exchanges the even (e) and odd (o) sublattice index of the atomic variables $\sigma^\alpha$ with $\alpha = \{x, y, z\}$ in the Hamiltonians in Eqs. (1-2). As described further in the caption of Fig. 2, the AFM+SR phase has a curious feature, namely that it is split in two regions, that are delimited by a touching point of two second-order phase transition lines of the SR$_{\text{UNI}}$ and the plain AFM phase at $V/|\Delta| = 1/2$. For this special value, the effective magnetic field on one sublattice vanishes. At this point, however, the transition becomes discontinuous. Direct, continuous transitions between phases with different broken symmetries are rare and sometimes accompanied by deconfined quantum criticality, as in the ground state of frustrated quantum spin models, for example [56].

## 1.3 Result 2: Even-odd sublattice peak in cavity spectrum

A convenient feature of these quantum-optical quantum simulators is the ability to perform non-destructive measurements of the dynamics via cavity spectra. For the moving atomic quantum gas and the associated polaritonic density excitations, such measurements have lead to fruitful experiment-theory comparison [57–61].

Here we show that the translation-symmetry breaking induced by the nearest-neighbor Rydberg-dressed interaction $V$ leads to a novel collective mode and peak in the spectrum. Figure 3 shows the cavity spectrum upon increasing $g$ from the (AFM+SR) phase with broken $T_{\text{lat}}$ symmetry into the SR$_{\text{UNI}}$ phase where translation symmetry is restored. We observe that the conventional normal-mode polariton picture first seen by Rempe and Kimble [62] –which is, with modifications, also applicable to the Dicke model [63]– becomes insufficient to describe the photon dynamics. By contrast, the even-odd mode softens in a specific way: Denoting the frequency of the polariton pole as $\nu$, we observe that both, its real part (gap) and the imaginary part (damping rate) vanish with coupling constant (from the AFM+SR phase toward line 2 in Fig. 2) with

$$\text{Re}[\nu] \propto \pm\sqrt{g_c - g}, \quad \text{Im}[\nu] \propto -(g_c - g) , \tag{7}$$

where $g_c$ refers to the right boundary delimiting the AFM+SR phase. This is because the translational symmetry $T_{\text{lat}}$ affects the atomic sector only which does not couple directly to the photonic rate of dissipation $\kappa$. This is in contrast to Dicke-type models or superfluids out-of-equilibrium (see [64] and [65]) whose dynamics becomes purely overdamped/imaginary at the transition [63], that is, the real part of the mode vanishes first.

## 1.4 Result 3: Photon number oscillations

We now account for a non-zero rate of atomic spontaneous emission $\gamma \neq 0$. Specific details of a given quantum-optical implementation (see Sec. 2) will determine which set of Lindblad

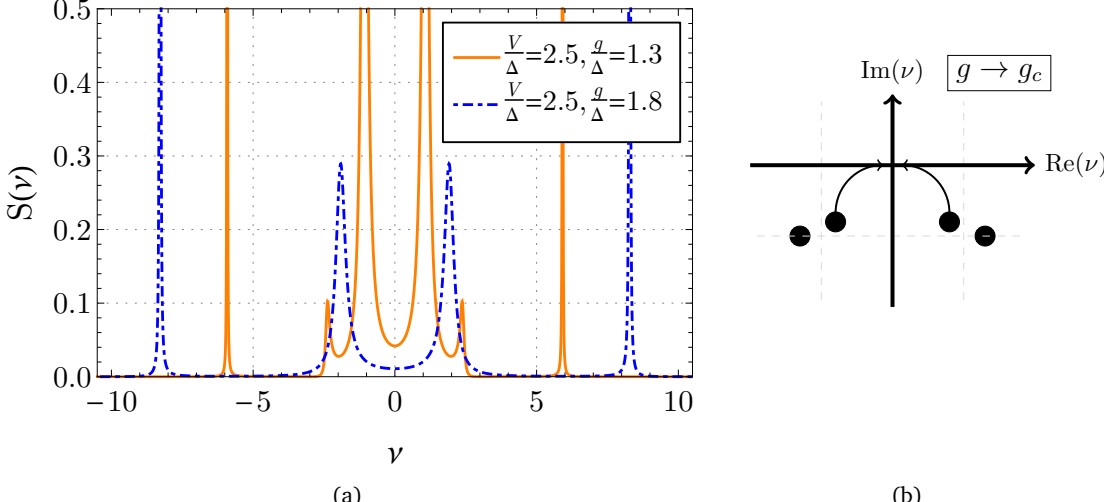

Figure 3: Even-odd sublattice peak in the cavity spectrum (peak close to zero frequency of the orange, solid line), which appears when the translational lattice symmetry $T_{\text{lat}}$ is spontaneously broken by the Rydberg-dressed nearest-neighbor interaction $V$. The two cavity spectra are computed for the positions labeled by (x) in Fig. 2. The blue, dashed line has the two polariton peaks in the uniform $\text{SR}_{\text{UNI}}$ phase with the photonic branch around the cavity frequency $\omega_0/\Delta = 2.0$. The orange, solid line is the spectrum in the AFM+SR regime with broken $T_{\text{lat}}$; it shows the prominent even-odd peak, which becomes soft toward the phase boundary (the right edge of the yellow strip in Fig. 2. (b) Low-energy pole structure of the even-odd sublattice peak, where both the real and the imaginary part of the poles vanish simultaneously as $g \to g_c$ according to Eq. (7).

operators and additional atomic levels need to be accounted for. In order to gain a first qualitative picture, we model an *effective* decay rate with $\mathscr{L}_\gamma[\rho]$ in Eq. (3) between the effective spin-up and spin-down states ($|1\rangle$ and $|0\rangle$) in Fig. 7. We expect $\gamma$ to become larger once the detuning to the shorter lived excited states is decreased; it is generally true that the effective ground state levels inherit a finite lifetime from admixing a short-lived state. For a specific experimental set-up, one may also include other types of atomic losses or dephasing.

This at first sight innocuous change has interesting consequences. Even qualitative features of Fig. 2 are drastically changed (although experimentally, for far enough detuned intermediate, excited states and rapid, enough ramps of the spin-light coupling $g$ these $\gamma$-induced changes may not be immediately visible). Allowing for a small $\gamma$, see Fig. 4, in particular wipes out the stable AFM phase and introduces a fully downward polarized state $\text{FP}_\downarrow$ as well as a novel oscillatory phase (AFM+SR)-OSC. Here also the photon field amplitude oscillates which can be detected by time-resolved measurements of the intensity of the light leaving the cavity.

At the root of these effect is $\mathscr{L}_\gamma[\rho]$ in Eq. (3): it explicitly breaks the discrete symmetry $G$ given by the product of time-reversal: $\mathscr{T}_\ell = -i\sigma_\ell^y K_\ell$, $t \to -t$ (for a spin $s = 1/2$) and spin rotation around the y-axis by $\pi$: $D_{y,\pi}^{1/2,\ell} = -i\sigma_\ell^y$. Here $K_\ell$ is the complex conjugation operator such that $\mathscr{G}_\ell = D_{y,\pi}^{1/2,\ell} \mathscr{T}_\ell = -K_\ell$. If we write $G = \Pi_{\ell=1}^N \mathscr{G}_\ell$ we have $GHG^{-1} = H$. In particular, this implies for steady states $\langle \sigma_{e,o}^y \rangle = \langle G\sigma_{e,o}^y G^{-1} \rangle = -\langle \sigma_{e,o}^y \rangle \overset{!}{=} 0$ if $\gamma = 0$. For $\gamma \neq 0$, the spins can start developing expectation values also in the $y$-direction. This offers new possibilities for the spin dynamics such as anomalous spin precession [27] not available in equilibrium.

We show the phases for a further range of parameters ($\Delta/\gamma$, $g/\gamma$) space for a fixed strength $V$ of the Rydberg interaction in two spatial dimensions in Fig. 5. In mean-field theory we

SciPost Phys. 1(1), 004 (2016)

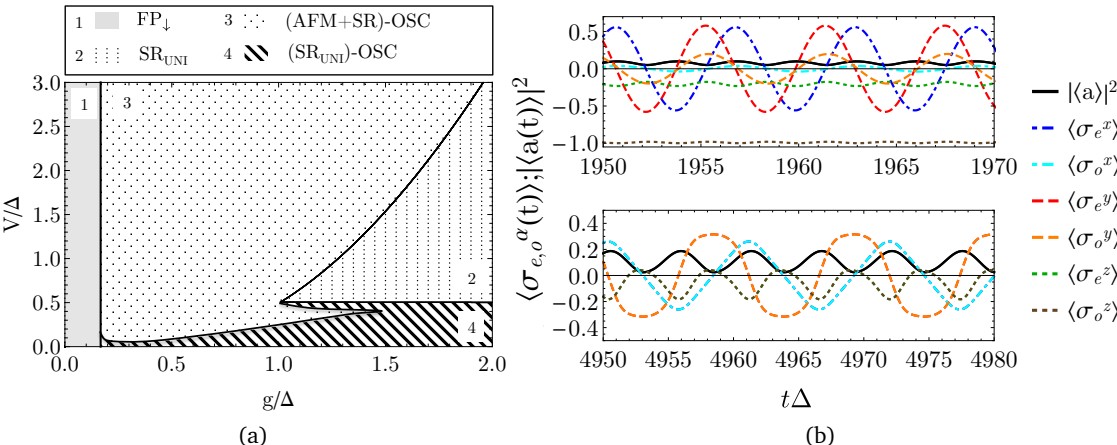

(a)                                                             (b)

Figure 4: (a) Supplementing the phase diagram Fig. 2 in the (V/$\Delta$,g/$\Delta$)-plane by a small amount of atomic dissipation $\gamma/\Delta = 0.01$ with ($\omega_0/\Delta = 2.0, \kappa/\Delta = 0.2$) yields a different picture. In comparison to the $\gamma/\Delta = 0.0$ case (compare Fig. 2), there are no stable steady-states with a broken lattice symmetry $T_{\text{lat}}$ any more. Instead, the system can show persistent oscillations in time. (b) Depiction of the non-uniform (top, (AFM+SR)-OSC) oscillations and the uniform (bottom, $SR_{\text{UNI}}$-OSC) oscillations that characterize the long time limit behavior of Eqs. (12-15) with finite $\gamma$. Parameters for the upper plot are ($V/\Delta = g/\Delta = 0.5$), the lower plot is obtained at ($V/\Delta = 0.3, g/\Delta = 1.2$).

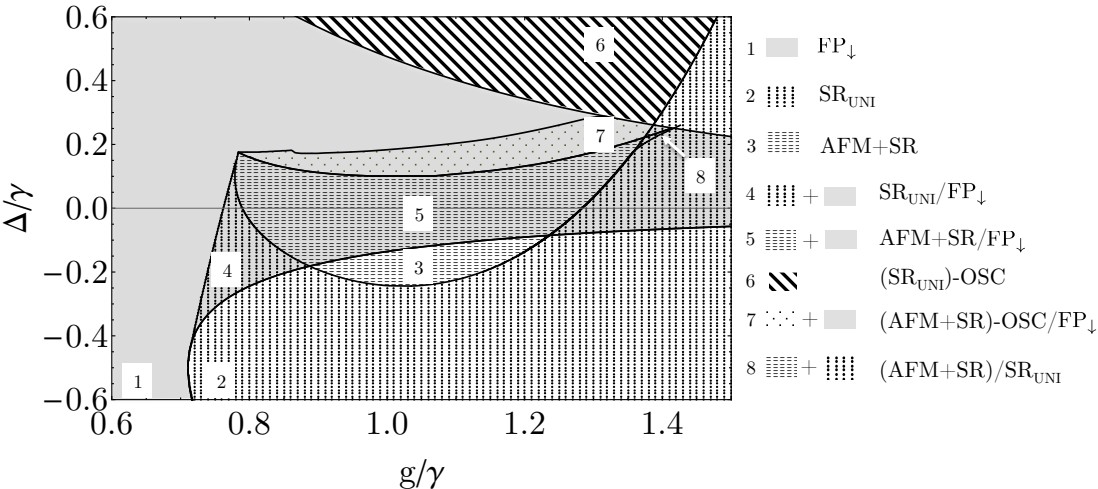

Figure 5: Non-equilibrium steady-state phase diagram of Eqs. (1-4) with finite, atomic spontaneous emission ($\kappa/\gamma = 0.2, V/\gamma = 1.8, \omega_0/\gamma = 2.0$). Apart from time-independent states, the dynamics also realises limit cycles where atomic and photon components show persistent oscillations in time. Oscillations can be uniform or different on the even/odd sublattice, see Fig. 6. Depending on the initial configuration, the system can reach different long-time fixpoints. Bistabilities occur whenever two phases overlap (see legend). Crystalline antiferromagentic order only occurs together with superradiance (AFM+SR).

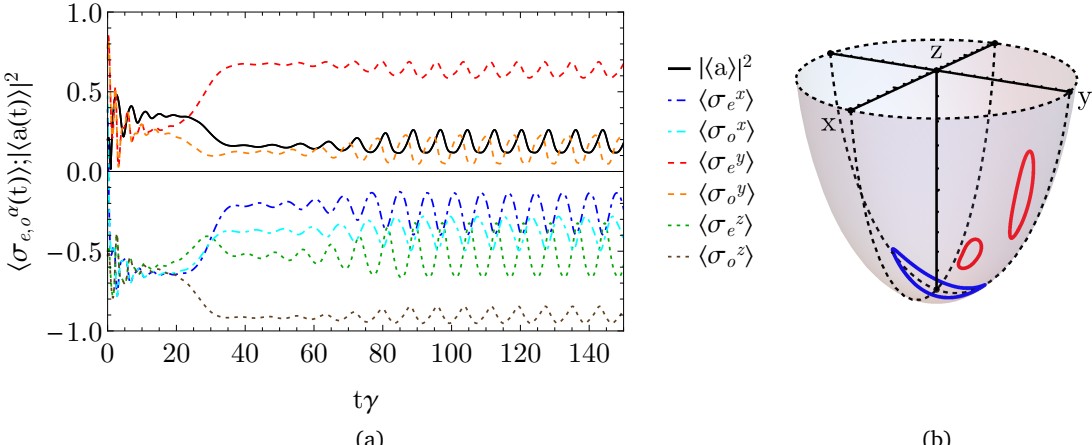

Figure 6: (a): Persistent spin and photon-field oscillations appear in the (AFM+SR)-OSC phase for $\Delta > 0$ close to the (AFM+SR) stable region for the parameters used in Fig. 5 for $g/\gamma = 1.1, \Delta/\gamma = 0.15$. Note that between $t\gamma \approx 10-25$ the amplitudes display (quasi-) plateaus followed by a rapid in/decrease toward their final mean values; and only then the oscillations begin and persist. For moving atomic gases in a cavity, we note that Ref. [66] found "pre-thermalized" plateaus in the time evolution for the order parameter by solving Fokker-Planck type kinetic equations. (b): Illustration of the atomic components for the limit cycles on the lower part of the Bloch sphere. The two upper lineshapes (red) depict the lines traced by the oscillations on the even and the odd sublattice in the (AFM+SR)-OSC phase, as depicted in (a). The lower lineshape illustrates a limit cycle of uniform oscillations of the atomic components in the $SR_{UNI}$ phase.

distinguish five phases in the long time limit. Three are steady-states denoted as ($FP_{\downarrow}$, $SR_{UNI}$, AFM+SR, see also Tab. 1) and two are stable limit cycles. The Lindblad operators try to drive the system into an empty state without any excitations; consequently AFM order can only occur in the presence of a coherent drive, i.e. together with a photon condensate $\langle a \rangle$. In the latter phases, the system exhibits oscillations in both atomic and photonic components, since the atomic dynamics couples back to the photon sector. The oscillations can be uniform in all components $(SR_{UNI})-OSC$ or different on the sublattices $(AFM + SR)-OSC$.

One point of view to interpret the oscillations data in Fig. 6 is that the (collective) spin oscillator (see also Ref. [67]) and the cavity oscillator synchronize with each other after a sufficient amount of time (see also Ref. [68]). Additionally, the system can show bistabilities, meaning that the eventual fate of the system in the long-time limit depends on the initial conditions. However, we also find a small strip in the phase diagram where the system is bistable between AFM + SR and a $SR_{UNI}$ phase. We will describe the phases more in Sec 4.3.

# 2 Ideas for quantum-optical implementation of the model

We seek an implementation which realizes the Hamiltonian in Eqs. (1,2). At the core of our model is the "two dipoles in-one" unit depicted in Fig. 7. As described in the caption, Dipole 1 could be created by weakly admixing a relatively low quantum number Rydberg level ($n \sim 30$) to a set of long-lived hyperfine split states $|1\rangle$ and $|0\rangle$. Dipole 2 couples to the cavity via two far-detuned, excited states $|d\rangle, |e\rangle$.

The atomic levels we consider to realize an effective spin system could be the hyperfine-structure manifold of the ground states of $^{87}$Rb. Typically this is the $5^2S_{1/2}$ state split into

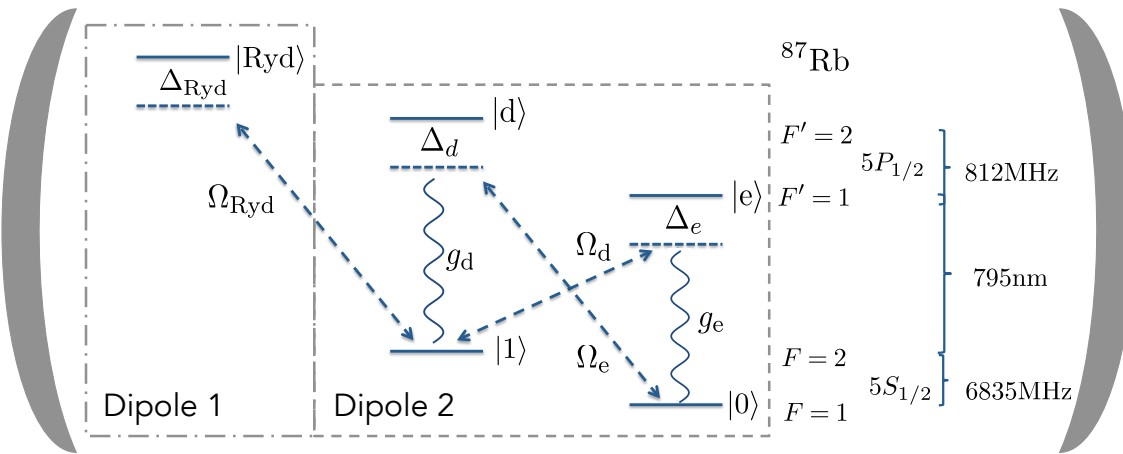

Figure 7: Blueprint for the wanted "two dipoles in-one" unit and the Hamiltonian Eqs. (1,2), which allows two independent photonic force carriers to couple to the atomic spin. The effective spin degree of freedom is encoded in the levels $|1\rangle$ and $|0\rangle$. The resonator mirrors (grey shades) confine the optical photon mode, which couples to the effective spin via $g_d$ and $g_e$. *Dipole 1*: A dressing laser can weakly admix a Rydberg level to a ground-state $|1\rangle \rightarrow |1\rangle + \frac{\Omega_{\mathrm{Ryd}}}{2\Delta_{\mathrm{Ryd}}} |\mathrm{Ryd}\rangle = |\tilde{1}\rangle$. We want the resulting effective potential $V_{\ell m}^{\mathrm{eff}}$ between $|\tilde{1}\rangle_\ell$ and $|\tilde{1}\rangle_m$ states to be predominantly nearest-neighbor in a square lattice spaced by an optical wavelength. Complex potentials including angle-dependencies can be realized [5, 24, 25]. *Dipole 2*: The double Raman scheme [47] provides a tunable coupling $\sim (a + a^\dagger)\sigma_\ell^x$ to the optical cavity. Choosing the lattice and cavity modefunction as in Fig. 1 results in a homogeneous coupling $g$, that is, all the spins couple in the same way to the cavity. This provides the infinite-range coupling between all the spins. The depicted level scheme corresponds to the ground state manifold $5S_{1/2}$ and the first excited state manifold $5P_{1/2}$ of $^{87}$Rb including their typical frequency splittings.

the $F = 1$ and the $F = 2$ manifold such that $|0\rangle = |\downarrow\rangle = |F = 1, m_F = -1\rangle$ and $|1\rangle = |\uparrow\rangle = |F = 2, m_F = -2\rangle$. Here, cavity-assisted Raman transitions couple the $(|0\rangle, |1\rangle)$ ground-states via adiabatic elimination of the detuned excited states $(|d\rangle, |e\rangle)$ to the cavity [47, 52]. Then, the cavity is (indirectly) pumped with photons from the transversal pumping-laser that scatter off the atoms and populate the cavity mode. In that way, the pump is "hidden" in the atom-photon coupling $g$: it is the counter-rotating terms that stabilize non-trivial steady-states with finite excitation number.

The second part of the pumping scheme, denoted as Dipole 1 in Fig. 7, consists of admixing a small part of a Rydberg state to the state $|1\rangle$ that is also coupled to the cavity. To first-order in perturbation theory of the driving, the ground-state becomes dressed with a small fraction of the excited state $|\tilde{1}\rangle \approx |1\rangle + \frac{\Omega_{\mathrm{Ryd}}}{2\Delta_{\mathrm{Ryd}}} |\mathrm{Ryd}\rangle + \mathcal{O}\left(\frac{\Omega_{\mathrm{Ryd}}}{2\Delta_{\mathrm{Ryd}}}\right)^2$, where $\Omega_{\mathrm{Ryd}}$ is the Rabi-frequency and $\Delta_{\mathrm{Ryd}}$ is the detuning from the Rydberg level. Ground-states $\{|\tilde{1}\rangle_i, |\tilde{1}\rangle_j\}$ interact then with a dressed Rydberg interaction that can be controlled by changing $(\Omega_{\mathrm{Ryd}}, \Delta_{\mathrm{Ryd}})$ of the dressing laser [69–72]. Dressing schemes for Rydberg atoms on optical lattices have recently been experimentally realised [5] and the effective Rydberg potential depends strongly on the chosen Rydberg states [24]. An additional degree of freedom that allows for engineering anisotropic effective potentials with an angular dependence $V_{ij}^{\mathrm{eff}}(r_{ij}, \theta_{ij})$ can be introduced by employing states with angular momentum such as states from $P$-manifolds [24].

In a suitably chosen rotating frame of reference, derived in Appendix B, the parameters appearing in the Hamiltonian Eqs. (1,2), can be expressed as follows. For the spin longitudinal

field $\Delta$ and the effective cavity frequency $\omega_0$, we have

$$\Delta = -\Delta_1 + \frac{\Omega_{\mathrm{Ryd}}^2}{4\Delta_{\mathrm{Ryd}}}\,, \qquad \omega_0 = N\frac{g_e^2}{\Delta_e} + \omega_a\,, \tag{8}$$

with $\omega_a$ and $\Delta_1$ defined in Eq. (66). The coherent coupling to the cavity can be tuned by the two external lasers $\Omega_{d,e}$:

$$g = \sqrt{N}\frac{g_{d,e}\Omega_{d,e}}{2\Delta_{d,e}}, \quad \frac{\Omega_d}{4\Delta_d} = \frac{\Omega_e}{4\Delta_e}, \quad \frac{g_d^2}{\Delta_d} = \frac{g_e^2}{\Delta_e}\,. \tag{9}$$

Finally, the Rydberg-mediated potential takes the general form [71]

$$V_{\ell m}^{\mathrm{eff}} = \left(\frac{\Omega_{\mathrm{Ryd}}}{2\Delta_{\mathrm{Ryd}}}\right)^4 \frac{C_6}{r_{\ell m}^6 + R_c^6}\,, \tag{10}$$

and the $V$ appearing in Eq. (1) evaluates Eq. (10) at a fixed $r_{\ell m}$ equal to the distance between neighboring lattice sites. Effectively step-like potentials are also possible. $R_c$ is the critical radius defined by $2\Delta_{\mathrm{Ryd}} \equiv V(R_c)$ which yields $R_c = \left(\frac{C_6}{2\hbar|\Delta_{\mathrm{Ryd}}|}\right)^{1/6}$. At smaller distances $r_{\ell m} < R_c$, dressing to doubly excited states becomes ineffective because of the large detuning $|\Delta_{\mathrm{Ryd}}| + V_{\ell m}$. The soft-core potentials contain a number of additional resonances at $r_{ij} \ll R_c$, which are undesirable to realise clean interactions. To ensure the interacting atoms in an optical lattice interact via the clean van-der-Waals tail, it is more advantageous to address relatively low-lying Rydberg states with principal quantum numbers $n \sim 30$, as $R_c$ can then also shrink down to optical wavelengths. Additionally, we comment that complementary to optical lattices, two-dimensional arrays of microtraps have already been used [73] to trap single $^{87}$Rb atoms with lattice spacings $\sim \mu m$. This would allow to use more highly excited Rydberg states for the dressing interaction bringing with it the advantage of longer lifetimes of higher lying Rydberg states.

In App. D, we present an overview of the relevant energy and time scales including a discussion on problematic Rydberg decays.

## 3 Coupled even-odd sublattice mean-field master equations for atoms and photons

We now derive and solve the coupled mean-field master equations for both, the spin degrees of freedom and the photons for an infinite number of atomic spins $N \to \infty$. In absence of the short-range, nearest-neighbor interaction $V$, a mean-field ansatz actually represents the exact solution in the long time limit $t \to \infty$ [53]. We account for different spin expectation values on the even versus odd sublattice of the bipartite square lattice of Fig. 1. The goal is to allow for steady-states with spontaneously broken translational (even-odd interchange) symmetry.

To this end, we now approximate the solution of the full master equation

$$\partial_t \rho = -i[H,\rho] + \mathcal{L}_\kappa[\rho] + \mathcal{L}_\gamma[\rho]\,, \tag{11}$$

by factorizing the spin part of density operator for all even sites as $\rho_e = \bigotimes_{\ell=1}^{N/2} \rho_{e,\ell}$ and analogously for the odd sites $\rho_o = \bigotimes_{\ell=1}^{N/2} \rho_{o,\ell}$. In the conclusions, Sec. 5, we comment further on the prospects of capturing finite spatial correlations and fluctuations beyond mean-field. We further define the spin expectation values on the even and odd sublattices, respectively:

$\langle \sigma_{e/o}^{\alpha} \rangle = \mathrm{Tr}[\rho_{e/o} \sigma_{e/o}^{\alpha}]$ where $\alpha$ refers to $(x, y, z)$. These six real-valued spin projections are complemented by two variables for the photon fields, $\langle a \rangle$, $\langle a^{\dagger} \rangle$, making it a total of eight variables to keep track of. This way, $\rho$ also includes entries to discriminate between the vacuum and a coherent photon field. The first four equations read

$$\partial_t \langle \sigma_e^x(t) \rangle = \langle \sigma_e^y(t) \rangle \left[ \Delta - 2V(\langle \sigma_o^z(t) \rangle + 1) \right] - \frac{\gamma}{2} \langle \sigma_e^x(t) \rangle \tag{12}$$

$$\partial_t \langle \sigma_e^y(t) \rangle = \langle \sigma_e^x(t) \rangle \left[ 2V(\langle \sigma_o^z(t) \rangle + 1) - \Delta \right] - 2g[\langle a(t) \rangle + \langle a^{\dagger}(t) \rangle] \langle \sigma_e^z(t) \rangle - \frac{\gamma}{2} \langle \sigma_e^y(t) \rangle \tag{13}$$

$$\partial_t \langle \sigma_e^z(t) \rangle = 2g[\langle a(t) \rangle + \langle a^{\dagger}(t) \rangle] \langle \sigma_e^y(t) \rangle - \gamma(1 + \langle \sigma_e^z(t) \rangle) \tag{14}$$

$$\partial_t \langle a(t) \rangle = -(\kappa + i\omega_0) \langle a(t) \rangle - \frac{1}{2} i g(\langle \sigma_e^x(t) \rangle + \langle \sigma_o^x(t) \rangle) \tag{15}$$

The equations for the odd sublattice spin projections follow from Eqs. (12-14) by exchanging the sublattice index $e \leftrightarrow o$. The complex conjugate of Eq. (15) completes the set of eight coupled equations. Here, we rescaled the photonic variable with $a(t) \to \sqrt{N} \langle a(t) \rangle$. A steady-state is macroscopically occupied in the thermodynamic limit and one may also define $\langle \sigma_{e,o}^{\alpha} \rangle \equiv$

$\frac{1}{(N/2)} \sum_{\ell \in even,odd}^{N} \sigma_{\ell}^{\alpha}$.

Mean field master equations are often a first step to study driven-dissipative systems, see for example, Refs. [26, 27, 74–78] and Ref. [79] for a variety of contexts.

# 4 Derivation and detailed discussion of results

## 4.1 Result 1: Combination of superradiance and magnetic translation symmetry-breaking

### 4.1.1 Phase boundaries and order parameters with photon losses ($\kappa \neq 0, \gamma = 0$)

We first consider the case where the atoms do not decay spontaneously and the only loss-process is given by the Lindblad $\mathscr{L}_{\kappa}$, see Eq. (4). The Eqs. (12-15) with $\gamma = 0$ conserve in this case a pseudo-angular momentum

$$\langle \sigma_{e,o}^x \rangle^2 + \langle \sigma_{e,o}^z \rangle^2 = 1 \tag{16}$$

provided we start from a low-entropy initial state for the spins (as discussed above in Subsec. 1.2), which fulfill this condition. Here $(e, o)$ refers to the even and odd sub lattice respectively. Due to the presence of time-reversal symmetry in the atomic channel, the steady-state of the system constrains $\langle \sigma_{e,o}^y \rangle = 0$ (see the discussion above in Sec. 1.4). Fig. 2 is calculated, with photon losses, by numerically solving for the stationary states of Eqs. (12-15). We determine stability by inspecting the real parts of the eigenvalues from the corresponding stability matrix that is obtained by linearising Eqs. (12-15) to first oder in fluctuations around the steady-states.

Together with the constraint Eq. (16), the set of Eqs. (12-15) can be solved analytically. The homogeneous ($\langle \sigma^{x,z} \rangle \equiv \langle \sigma_e^{x,z} \rangle = \langle \sigma_o^{x,z} \rangle$) steady-state solutions, which describe the Dicke

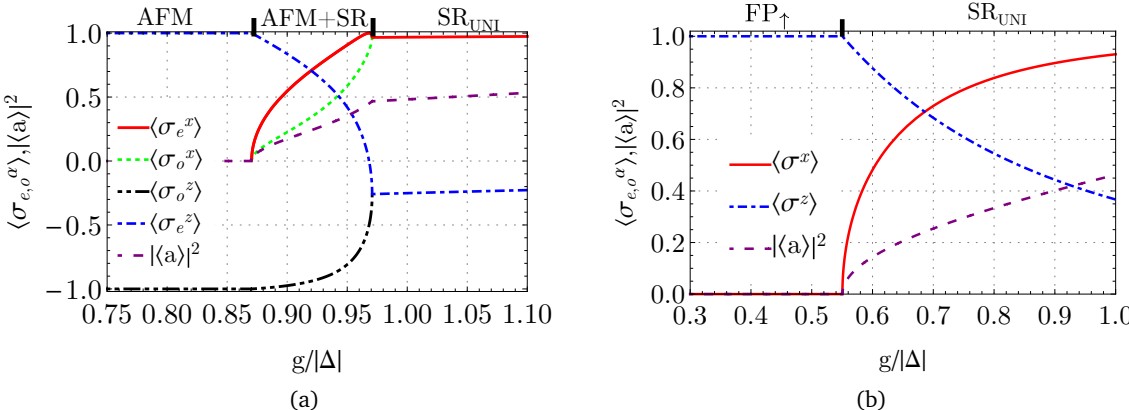

Figure 8: Steady-states for Eqs. (12-15) with $\gamma = 0$ for the parameters $\omega_0 = 2.0|\Delta|, \kappa = 0.2|\Delta|$ for the phases depicted in Fig. 2. (a) Behavior of the magnetisations and the coherent photon condensate in the plain antiferromagnet (AFM), in the regime where translational symmetry breaking and a superradiant photon condensate occur together (AFM+SR) and in the Dicke-phase ($SR_{UNI}$) as the atom-light coupling $g$ is increased at a fixed value of $V = |\Delta|$. All transitions are continuous in the order-parameters. The plot in (b) shows the onset of superradiance as the atom-light coupling is increased, for a fixed value of $V = 0.1|\Delta|$.

superradiance transition are

$$\langle a \rangle = \mp \frac{g \langle \sigma^x \rangle}{\omega_0 - i\kappa}, \tag{17}$$

$$\langle \sigma^x \rangle = \mp \frac{\sqrt{J - J_c}\sqrt{4J + \Delta}}{\sqrt{2J + V}}, \tag{18}$$

$$\langle \sigma^z \rangle = \frac{J_c + \frac{\Delta}{4}}{2J - J_c + \frac{\Delta}{4}}. \tag{19}$$

We have defined $J = \frac{g^2 \omega_0}{\kappa^2 + \omega_0^2}$ and $J_c = \frac{\Delta}{4} - V$. A plot of the Eqs. (17)-(19) is illustrated in Fig. 8b. Starting in the antiferromagnetic phase and increasing the atom-light coupling $g$, a regime where superradiance ($\langle a \rangle \neq 0$) and a phase with different sub-lattice magnetisations occur together is predicted ($\langle \sigma_e^z \rangle - \langle \sigma_o^z \rangle \neq 0$) which, due to Eq. (16) implies $\langle \sigma_e^x \rangle - \langle \sigma_o^x \rangle \neq 0$. Due to the finite longitudinal field $\Delta$, one sublattice is easier flipped than the other and the system realizes a "canted" antiferromagnet. If the atom-light coupling is increased even further, translational symmetry is restored and the system realises a Dicke superradiant phase. This can be seen by tracking the evolution of the magnetisation as $g$ is increased in Fig. 8a.

We now derive analytical expressions for the phase-transition lines. First, we transform the mean-field equations (12-15) in frequency space via Fourier transformation

$$\mathcal{O}(t) = \frac{1}{\sqrt{2\pi}} \int_{-\infty}^{\infty} e^{-i\nu t} \mathcal{O}(\nu) d\nu, \quad \mathcal{O}^\dagger(t) = \frac{1}{\sqrt{2\pi}} \int_{-\infty}^{\infty} e^{-i\nu t} \mathcal{O}^\dagger(-\nu) d\nu \tag{20}$$

In general, one should add Markovian quantum noise-operators with zero-mean to the photonic and atomic set of the mean-field master equations that result from the interaction of the atom-cavity system with the vacuum modes outside of the cavity. These we denote $\mathscr{F}_{e,o}^\alpha(\nu)$ as the atomic and $\mathscr{F}^a(\nu)$ as the photonic noise-operators in frequency space.

Next, we add back fluctuations to Eqs. (12-15)

$$\langle \sigma_{e,o}^\alpha(\nu) \rangle \to \langle \sigma_{e,o}^\alpha \rangle \delta(\nu)\sqrt{2\pi} + \delta\sigma_{e,o}^\alpha(\nu) \tag{21}$$

$$\sqrt{N}\langle a(\nu) \rangle \to \sqrt{N}\langle a \rangle \delta(\nu)\sqrt{2\pi} + \delta a(\nu), \tag{22}$$

where the steady-states are denoted as $\langle \sigma_{e,o}^\alpha \rangle$ with $\alpha = (x, y, z)$ and $\langle a \rangle$ is the expectation value for a coherent photon condensate. Here, $\delta\sigma_{e,o}^\alpha(\nu)$ and $\delta a(\nu)$ describe quantum fluctuations about the semi-classical steady-state and $\delta(\nu)$ denotes a delta function in frequency space. At long times, we may neglect second-order terms in the fluctuations by assuming that the steady-state values are large compared to the associated fluctuations in the thermodynamic limit $N \to \infty$.

The now linearized equations can be cast in matrix form.

$$\mathscr{F}(\nu) = \delta(\nu)f(\boldsymbol{\sigma}) + \boldsymbol{G}_R^{-1}(\nu) \cdot \boldsymbol{\delta\sigma}(\nu) \tag{23}$$

where the fluctuations around the steady state are collected in $\boldsymbol{\delta\sigma}(\nu)$ and the noise-operators are collected into $\mathscr{F}(\nu)$:

$$\boldsymbol{\delta\sigma}^T(\nu) = \left( \delta\sigma_e^x(\nu), \delta\sigma_e^y(\nu), \delta\sigma_e^z(\nu), \delta a(\nu), \delta a^\dagger(-\nu), \delta\sigma_o^x(\nu), \delta\sigma_o^y(\nu), \delta\sigma_o^z(\nu) \right) \tag{24}$$

$$\mathscr{F}^T(\nu) = \left( \mathscr{F}_e^x(\nu), \mathscr{F}_e^y(\nu), \mathscr{F}_e^z(\nu), \mathscr{F}^a(\nu), \mathscr{F}^{a^\dagger}(-\nu), \mathscr{F}_o^x(\nu), \mathscr{F}_o^y(\nu), \mathscr{F}_o^z(\nu) \right) \tag{25}$$

The function $f(\boldsymbol{\sigma})$ is associated with the coherent part of the steady-states and thus only leads to a zero-frequency peak in the cavity-spectrum. The responses of the fluctuations $\boldsymbol{\delta\sigma}$ to the noise or 'driving-forces' $\mathscr{F}$ is encoded by the retarded Green-function $\boldsymbol{G}_R(\nu)$, its inverse $\boldsymbol{G}_R^{-1}(\nu)$ is given as:

$$\begin{pmatrix}
\frac{1}{2}(\gamma - 2i\nu) & 2V(\langle\sigma_o^z\rangle + 1) - \Delta & 0 & 0 & 0 & 0 & 0 & 2V\langle\sigma_e^y\rangle \\
\Delta - 2V(\langle\sigma_o^z\rangle + 1) & \frac{1}{2}(\gamma - 2i\nu) & 2g(\langle a\rangle + \langle a^\dagger\rangle) & 2g\langle\sigma_e^z\rangle & 2g\langle\sigma_e^z\rangle & 0 & 0 & -2V\langle\sigma_e^x\rangle \\
0 & -2g(\langle a\rangle + \langle a^\dagger\rangle) & \gamma - i\nu & -2g\langle\sigma_e^y\rangle & -2g\langle\sigma_e^y\rangle & 0 & 0 & 0 \\
\frac{1}{2}ig & 0 & 0 & \kappa - i(\nu - \omega_0) & 0 & \frac{1}{2}ig & 0 & 0 \\
-\frac{1}{2}ig & 0 & 0 & 0 & \kappa - i(\nu + \omega_0) & -\frac{1}{2}ig & 0 & 0 \\
0 & 0 & 2V\langle\sigma_o^y\rangle & 0 & 0 & \frac{1}{2}(\gamma - 2i\nu) & 2V(\langle\sigma_e^z\rangle + 1) - \Delta & 0 \\
0 & 0 & -2V\langle\sigma_o^x\rangle & 2g\langle\sigma_o^z\rangle & 2g\langle\sigma_o^z\rangle & \Delta - 2V(\langle\sigma_e^z\rangle + 1) & \frac{1}{2}(\gamma - 2i\nu) & 2g(\langle a\rangle + \langle a^\dagger\rangle) \\
0 & 0 & 0 & -2g\langle\sigma_o^y\rangle & -2g\langle\sigma_o^y\rangle & 0 & -2g(\langle a\rangle + \langle a^\dagger\rangle) & \gamma - i\nu
\end{pmatrix} \tag{26}$$

The frequency-resolved spectrum of excitations governed by the fluctuations can be obtained from the characteristic equation

$$\mathrm{Det}[\boldsymbol{G}_R^{-1}(\nu)] = 0. \tag{27}$$

All poles of the retarded Green function are located in the lower complex frequency plane. The damping rate of the excitations can be read off from the imaginary part of these poles, see for instance Reference [80]. In the case of second order transitions, the order-parameters ($\langle a \rangle$, $\langle \sigma_{e,o}^x \rangle$, $\langle \sigma_{e,o}^z \rangle$) change continuously at the phase transitions. We obtain analytical expressions for the phase boundaries by solving

$$\lim_{\nu \to 0} \mathrm{Det}[\boldsymbol{G}_R^{-1}(\nu)] = \alpha^2 = 0. \tag{28}$$

The zeroth-order frequency component refers to a possible gap $\alpha^2$ of the system that will close continuously ($\lim_{g \to g_c} \alpha^2 \to 0$) when the phase transition is approached by increasing the atom-light coupling $g$. We arrive at the set of transition lines given by Eq. (30), Eq. (31) and Eq. (33) that are depicted as black lines in Fig. 2 where they match the numerically calculated transitions. The open character of the system becomes manifest in the expressions for the phase boundaries as all transitions explicitly depend on the rate of photonic dissipation $\kappa$.

Starting from the $FP_\uparrow$ phase, the Dicke superradiance transition in presence of the Rydberg interaction sets in at the critical coupling strength:

$$g_{c,1} = \frac{\sqrt{\kappa^2 + \omega_0^2}\sqrt{\omega_0(\Delta - 4V)}}{2\omega_0}, \tag{29}$$

$$V_{c,1} = \frac{\Delta}{4} - \frac{g^2\omega_0}{\kappa^2 + \omega_0^2}. \tag{30}$$

The finite Rydberg-dressed interaction $V$ modifies the effective longitudinal field experienced by the spins which shifts the position of the superradiant condensate in comparison to the $V = 0$ case. Eq. (29) collapses to the familiar Dicke superradiance transition in the case $V \to 0$ [47]. The crossover from the (AFM+SR) regime to the Dicke superradiant phase (SR) is given by:

$$V_{c,2} = \left[ 4g^2\omega_0 \left( 4g^2\omega_0 + \sqrt{\Delta^2 \left(\kappa^2 + \omega_0^2\right)^2 - 8\Delta g^2\omega_0 \left(\kappa^2 + \omega_0^2\right) + 80g^4\omega_0^2} \right) \right.$$
$$+ \Delta \left(\kappa^2 + \omega_0^2\right) \sqrt{\Delta^2 \left(\kappa^2 + \omega_0^2\right)^2 - 8\Delta g^2\omega_0 \left(\kappa^2 + \omega_0^2\right) + 80g^4\omega_0^2}$$
$$\left. + \Delta^2 \left(\kappa^2 + \omega_0^2\right)^2 \right] / 8 \left(\kappa^2 + \omega_0^2\right)\left(\Delta \left(\kappa^2 + \omega_0^2\right) + 2g^2\omega_0\right). \tag{31}$$

The transition line from the AFM phase to the AFM+SR regime is given by $(-\Delta < 0)$

$$g_{c,3} = \frac{\sqrt{\Delta}\sqrt{\kappa^2 + \omega_0^2}\sqrt{4V - \Delta}}{2\sqrt{2}\sqrt{V}\sqrt{\omega_0}}, \tag{32}$$

$$V_{c,3} = \frac{\Delta^2 \left(\kappa^2 + \omega_0^2\right)}{4\left(\Delta \left(\kappa^2 + \omega_0^2\right) - 2g^2\omega_0\right)}. \tag{33}$$

We note that the line where AFM and SR order occur together diverges $V_{c,3} \to \infty$ as $\lim_{g \to g_\star}$ with $g_\star = \left(\sqrt{\Delta}\sqrt{\kappa^2 + \omega_0^2}\right)/\left(\sqrt{2}\sqrt{\omega_0}\right)$. Moreover, on a mean-field level, there is a touching point $g_t$ of two second-order phase transition lines that can be found by equating Eq. (31) and Eq. (33) which yields $g_t = \left(\sqrt{\Delta}\sqrt{\kappa^2 + \omega_0^2}\right)\left(2\sqrt{\omega_0}\right)$ and $V_{c,3}(g_t) = \Delta/2$ marks the point where the effective longitudinal field on one of the sublattices vanishes. On a mean-field level, we find a multi-critical point, where all second-order phase transition lines meet on the $g = 0$-axis at $V = \Delta/4$.

In Appendix A, we analyze a $T = 0$ equilibrium spin model with the same phases as Fig. 2 upon identifying one spin interaction constant with cavity parameters. Dynamics and statistics remains drastically different in the non-equilibrium case, however.

## 4.2 Result 2: Even-odd sublattice peak in cavity spectrum

### 4.2.1 Derivation of the cavity output spectrum ($\kappa \neq 0, \gamma = 0$)

Here we calculate the frequency-resolved cavity output spectrum for the light that leaks from the imperfect cavity mirrors within a standard input-output theory [47, 81, 82]. We find that every phase in Fig. 2 shows a characteristic cavity output spectrum making it possible to experimentally distinguish one phase from the other. The input fields are related to the output fields by the relation

$$a_{\text{out}}(\nu) = \sqrt{2\kappa}a(\nu) - a_{\text{in}}(\nu), \tag{34}$$

$$a_{\text{out}}^\dagger(-\nu) = \sqrt{2\kappa}a^\dagger(-\nu) - a_{\text{in}}^\dagger(-\nu). \tag{35}$$

The annihilation operators $(a_{\text{out}}(\nu), a_{\text{in}}(\nu), a(\nu))$ correspond to the output field, the vacuum input field, and the intra cavity field, respectively and we have used $\mathscr{F}_a(\nu) = \sqrt{2\kappa}a_{\text{in}}(\nu)$. The Markovian quantum noise operators with zero mean are determined by their second-order correlation functions. For the photonic channel they read

$$\langle a_{\text{in}}(\nu')a_{\text{in}}^\dagger(-\nu)\rangle = \delta(\nu+\nu') \tag{36}$$

We solve Eq. (23) for $a_{\text{out}}(\nu)$ and $a_{\text{out}}^\dagger(-\nu)$ together with Eq. (36) to obtain the output spectrum for a vacuum input field

$$S(\nu) = \langle a_{\text{out}}^\dagger(\nu)a_{\text{out}}(\nu)\rangle = 2\kappa\,\langle \delta a^\dagger(\nu)\delta a(\nu)\rangle = 2\kappa\int_{-\infty}^{\infty} e^{-i\nu\tau}\,\langle \delta a^\dagger(0)\delta a(\tau)\rangle\,d\tau. \tag{37}$$

The unnormalized fluorescence spectrum $S(\nu)$ is proportional to finding a photon at frequency $\nu$ and thus displays the position and the spectral weight of the resonance energies of hybridized atom-cavity modes. We only depict cavity-spectra in the $\gamma = 0$ limit. We have investigated the effect of spontaneous emission on the cavity spectra in the $V \to 0$ limit previously [53] and found that it can induce a frequency asymmetry in the cavity spectrum since atomic excitations can leave the cavity directly by emission into free space. The cavity spectra for $\kappa \neq 0, \gamma = 0$ can be obtained for every phase in Fig. 2. In the fully polarized phase ($\langle a\rangle = \langle\sigma^x\rangle = 0, \langle\sigma^z\rangle = 1$) for $g < g_{c,1}$ and $-\Delta < 0$, the cavity spectrum is obtained as

$$S_1(\nu) = \frac{16\kappa^2 g^4(\Delta-4V)^2}{|\Omega_1|^2}, \tag{38}$$

$$|\Omega_1|^2 = \big|\big((\kappa-i\nu)^2(-\Delta+\nu+4V)(\Delta+\nu-4V)+4g^2\omega_0(\Delta-4V)$$
$$+\omega_0^2(-\Delta+\nu+4V)(\Delta+\nu-4V)\big)\big|^2$$

and is depicted in Fig. 9c. In the limit $V \to 0$ it reduces to the familiar expression obtained in Ref. [47]. In the AFM phase ($\langle\sigma_e^z\rangle = -1, \langle\sigma_o^z\rangle = 1, \langle a\rangle = \langle\sigma^x\rangle = 0$), the spectrum is given by

$$S_3(\nu) = \frac{64\kappa^2 g^4 V^2\left(\Delta^2-4\Delta V+\omega^2\right)^2}{|\Omega_3|^2} \tag{39}$$

$$\Omega_3 = \big|8g^2V\omega_0\left(\Delta^2-4\Delta V+\omega^2\right)+(\omega-\Delta)(\Delta+\omega)(\kappa-i\omega)^2(-\Delta+4V+\omega)(\Delta-4V+\omega)$$
$$+\omega_0^2(\omega-\Delta)(\Delta+\omega)(-\Delta+4V+\omega)(\Delta-4V+\omega)\big|^2 \tag{40}$$

and a typical spectrum can be seen in Fig. 9a. In the homogeneous phase we make use of Eqs. (17-19) and obtain the spectrum as

$$S_4(\nu) = \frac{16\kappa^2 g^8\omega_0^2\left(\kappa^2+\omega_0^2\right)^6(\Delta-2V)^4}{|\Omega_4|^2} \tag{41}$$

with the abbreviated expressions

$$\Omega_4 = \Big| 4g^4 \omega_0^2 (\Omega_{11} + \Omega_{42}) + V^2 (\kappa^2 + \omega_0^2)^2 (\Omega_{43} + \Omega_{44} + \Omega_{45})$$
$$+ 2g^2 V \omega_0 (\kappa^2 + \omega_0^2)(\Omega_{46} + \Omega_{47} + \Omega_{48}) \Big|^2,$$
$$\Omega_{41} = \Delta^2 (\kappa^2 + \omega_0^2)^3 + \kappa^6 \omega^2 + 2i\kappa^5 \omega^3 - \kappa^4 (\omega^4 - 3\omega^2 \omega_0^2)$$
$$+ 4i\kappa^3 \omega^3 \omega_0^2 + \omega_0^2 (\omega_0^2 - \omega^2)(\omega^2 \omega_0^2 - 16g^4),$$
$$\Omega_{42} = \kappa^2 \omega_0^2 (-16g^4 - 2\omega^4 + 3\omega^2 \omega_0^2) - 2i\kappa (16g^4 \omega \omega_0^2 - \omega^3 \omega_0^4),$$
$$\Omega_{43} = -\kappa^4 (8\Delta g^2 \omega_0 + \omega^4 - 3\omega^2 \omega_0^2) + 4i\kappa^3 \omega \omega_0 (\omega^2 \omega_0 - 4\Delta g^2)$$
$$+ 2i\kappa \omega \omega_0^2 (-8\Delta g^2 \omega_0 - 48g^4 + \omega^2 \omega_0^2),$$
$$\Omega_{44} = \kappa^6 \omega^2 + 2i\kappa^5 \omega^3 + \kappa^2 \omega_0 (8\Delta g^2 (\omega^2 - 2\omega_0^2) - 32g^4 \omega_0 - 2\omega^4 \omega_0 + 3\omega^2 \omega_0^3),$$
$$\Omega_{45} = \omega_0^2 (8\Delta g^2 \omega_0 (\omega^2 - \omega_0^2) + 16g^4 (3\omega^2 - 2\omega_0^2) - \omega^4 \omega_0^2 + \omega^2 \omega_0^4),$$
$$\Omega_{46} = -8\Delta g^2 \omega_0 (\kappa^2 + \omega_0^2)^2 + \Delta^2 (\kappa^2 + \omega_0^2)^2 (\kappa^2 + 2i\kappa \omega - \omega^2 + \omega_0^2),$$
$$\Omega_{47} = 2(\kappa^6 \omega^2 + 2i\kappa^5 \omega^3 - \kappa^4 (\omega^4 - 3\omega^2 \omega_0^2) + 4i\kappa^3 \omega^3 \omega_0^2 + \omega_0^2 (\omega_0^2 - \omega^2)(\omega^2 \omega_0^2 - 24g^4)),$$
$$\Omega_{48} = 2(\kappa^2 \omega_0^2 (-24g^4 - 2\omega^4 + 3\omega^2 \omega_0^2) - 2i\kappa (24g^4 \omega \omega_0^2 - \omega^3 \omega_0^4)),$$

depicted in Fig. 9d. We denote the cavity spectrum in the AFM+SR regime as $S_2(\nu)$. We solve Eqs. (12-15) numerically in the long-time limit and use Eq. (37) to determine the spectrum numerically, see Fig. 9b. Now we discuss the characteristic features and the behavior of the poles.

### 4.2.2 Discussion of cavity spectra and low-frequency pole structure for ($\kappa \neq 0, \gamma = 0$)

We begin our discussion with an analysis of the cavity spectra $S(\nu)$ in each phase. The cavity spectra are shown in Fig. 9. In general there are either four or six poles in the cavity spectrum. In the former case these originate from two photon-branches and two atomic branches that are symmetrically arranged around the zero-frequency axis. We identify the branches by their $g \to 0$ limit in the fully polarised phase where the resonances settle at the bare frequencies given by $\nu_{\text{Atom}} = \pm\Delta$ and $\nu_{\text{Photon}} = \pm\omega_0 - i\kappa$. There are six poles when the translational symmetry in the atomic sector is broken. The additional poles reflect the even/odd imbalance of the system and are thus attributed to the Rydberg interaction, see Fig. 9a. This provides a clear feature to experimentally detect a phase with antiferromagnetic order.

We describe and depict the characteristic features of the cavity spectra for each phase of the mean-field phase diagram in Fig. 2 below. The cavity spectra in Fig. 9c (fully polarized FP$_\uparrow$) and in Fig. 9d (superradiant phase) are well-known and derived in [47] in the $V \to 0$ limit. In the superradiant regime, an increasing Rydberg interaction $V$ shifts the atomic poles to higher energies whereas the peaks associated to the photonic branch settle around the cavity resonance at $\text{Re}(\nu) = \pm\omega_0 = \pm2.0|\Delta|$. In the AFM and the (AFM+SR) phase, the cavity spectra depicted in Fig. 9a and Fig. 9b exhibit the aforementioned even/odd sublattice peak that reflects the broken translational symmetry in the atomic sector, so that there are six poles in total.

The frequency-resolved eigenenergies of the hybridized atom-cavity modes display a characteristic behaviour close to the phase transition as $g \to g_{\text{c},1,2,3}$ in Fig. 2. On the real frequency axis, all phase transitions appear when at least one of the either four or six poles hits the zero $\nu = 0$. The low-frequency behaviour of the critical poles leading to the Dicke superradiance transition has already been established [83]. From the four poles, two (which we refer to as ($\nu_a, \nu_b$) in the following) approach the origin in the complex frequency plane. First, both poles become completely imaginary as $g \to g_{c,1}$. A single one of these poles vanishes at the

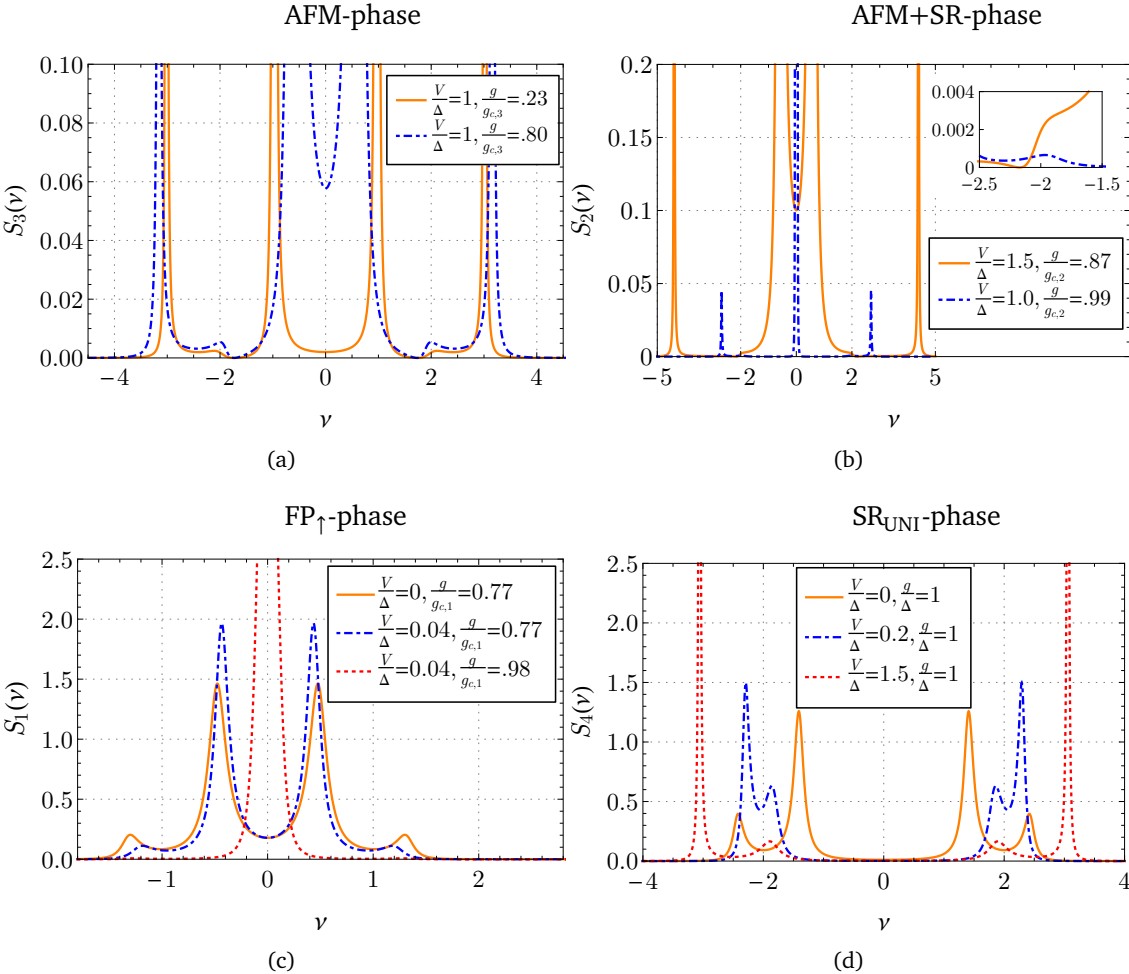

Figure 9: Typical cavity spectra for each of the four phases depicted in Fig. 2. In general, the resonances show the hybridised atom-cavity eigenenergies that can be obtained from solving Eq. (27). There are four poles in the cavity spectrum when translational symmetry is intact as in (c,d) and there are six poles when translational symmetry is broken as (a,b). (a) Cavity spectrum in the plain antiferromagnetic phase. The broken translational symmetry is reflected in the appearance of an additional (Rydberg) resonance in the atomic-sector. (b) Cavity spectra in the regime of a broken $\mathbb{Z}_2$ and translational symmetry $T$. As $g \to g_{c,2}$, two of the six poles move towards $\nu = 0$ (dot-dashed line). At $g = g_{c,2}$ translational symmetry is restored and the additional Rydberg-induced even-odd peaks disappear. (Inset) Close to the frequency at $\nu = \omega_0/\Delta = \pm 2.0$ there are resonances with small but finite weight corresponding to the cavity resonance. (c) Cavity spectrum for the fully polarized phase FP$_\uparrow$ with no photonic excitations $\langle a \rangle = 0$. (d) Spectrum in the superradiant regime $\langle a \rangle \neq 0$ with translational symmetry $T$ still intact.

phase-transition $|\nu_a| = 0$ while the other retains a finite imaginary part at $g = g_{c,1}$ set by the dissipation ($\nu_b(g = g_{c,1}) \sim -i\kappa$) emphasizing that the Dicke superradiance transition directly couples to the dissipation. The cavity spectrum in Fig. 9c exhibits a pole at zero frequency but with a finite, purely imaginary contribution. The intensity under this finite-width peak diverges, which indicates a macroscopic occupation of the cavity-mode $\langle a \rangle \neq 0$. As the transition from the AFM into the (AFM+SR)-phase involves a superradiance transition, the same behaviour of the low-frequency poles can be observed as $g \to g_{c,3}$. When translational symmetry is broken, this transition mainly affects the atomic channel. As the photonic sector alone

couples to a dissipative channel, we numerically observe that when translational symmetry is restored as one goes from the (AFM+SR) into the SR phase as $g \rightarrow g_{c,2}$, two of the six poles approach the origin on the complex frequency plane. In contrast to the Dicke superradiance transition both the real and imaginary part of the two low-frequency poles vanish together, see Fig. 3b for an illustration of the pole structure and Fig. 9b for the cavity spectra in the (AFM+SR) phase. At $g > g_{c,2}$ translational symmetry $T_{lat}$ is restored and the spectrum is given by Fig. 9d.

## 4.3 Result 3: Photon number oscillations

### 4.3.1 Phase boundaries and order parameters with spontaneous emission ($\kappa \neq 0, \gamma \neq 0$)

As discussed in Sec. 1.4, allowing for spontaneous emission ($\gamma \neq 0$) in addition to photon leakage ($\kappa \neq 0$) has a dramatic influence on the phase diagram obtained from the behaviour of the mean-field master equations in the long-time limit. In comparison to the $\gamma = 0$ case, the phase diagram is enriched by the presence of oscillatory and bistable phases, see Fig. 4 and Fig. 5. We first turn our attention to the case where there is a small amount of dissipation in the atomic channel ($\gamma = 0.01\Delta$) to analyse its impact on the long-time limit behaviour of the steady-state phases depicted in Fig. 2. We observe (see Fig. 4) that allowing for a small amount of dissipation, there are no stable steady-states that involve a broken lattice symmetry $T_{lat}$. The only steady-states in the investigated ($V/\Delta, g/\Delta$)-plane is the empty atom-cavity system ($FP_{\downarrow}$) and a uniform superradiant phase ($SR_{UNI}$). The remaining long-time limit behavior is characterized by persistent oscillations that can be uniform ($SR_{UNI} - OSC$) or non-uniform (($AFM + SR_{UNI}) - OSC$). As the Rydberg interaction is conditioned on population in the upper state, the phase boundary of the empty atom-cavity system is independent of $V$. Formally, we obtain its phase boundary by inspecting the eigenvalues of the stability matrix corresponding to the fixed point ($\langle a \rangle = \langle \sigma^x \rangle = \langle \sigma^y \rangle = 0, \langle \sigma^z \rangle = -1$). The real part of at least one of the associated stability eigenvalues becomes positive when

$$\gamma\kappa\left((\gamma + 2\kappa)^2 + 4\Delta^2\right)^2 - 32\Delta\lambda^2\omega_0(\gamma + 2\kappa)^2$$
$$+ 8\gamma\kappa\omega_0^2(\gamma - 2\Delta + 2\kappa)(\gamma + 2(\Delta + \kappa)) + 16\gamma\kappa\omega_0^4 = 0. \quad (42)$$

Solving for g yields,

$$g_{(crit,FP_{\downarrow})} = \frac{\sqrt{\gamma\kappa}\sqrt{((\gamma + 2\kappa)^2 + 4\Delta^2)^2 + 8\omega_0^2(\gamma - 2\Delta + 2\kappa)(\gamma + 2(\Delta + \kappa)) + 16\omega_0^4}}{4\sqrt{2}\sqrt{\Delta\omega_0(\gamma + 2\kappa)^2}}. \quad (43)$$

Here, we observe that at $g = g_{(crit,FP_{\downarrow})}$ the associated linearized stability matrix of Eqs. (12-15) obtains a pair of purely imaginary complex conjugated eigenvalues which signalizes that the system changes into a limit cycle via a Hopf bifurcation. Limit cycles in driven-dissipative models have been observed before e.g. in spin-1/2 systems [44, 84] and with Bose-Einstein condensates in optical cavities, see e.g. References [85, 86] and in driven QED-cavity arrays [87].

The transition into the stable $SR_{UNI}$-phase is discontinuous and we can solve Eqs. (12-15) in the long-time limit to obtain the photon condensate as

$$|\langle a \rangle|^2$$
$$= \frac{16J^3(2V - \Delta) \pm 2\sqrt{J^2(\Delta + 4J)^2(4J^2(\Delta - 2V)^2 - 2\gamma^2 JV - \gamma^2 V^2)} - 2J^2(\gamma^2 + 2\Delta(\Delta - 2V)) - \gamma^2 JV}{16J\omega_0(2J + V)^2}$$
$$= \frac{J^2(\Delta + 4J)(2V - \Delta) \pm \sqrt{J^4(\Delta + 4J)^2(\Delta - 2V)^2}}{4J\omega_0(2J + V)^2} + \mathcal{O}(\gamma^2), \quad (44)$$

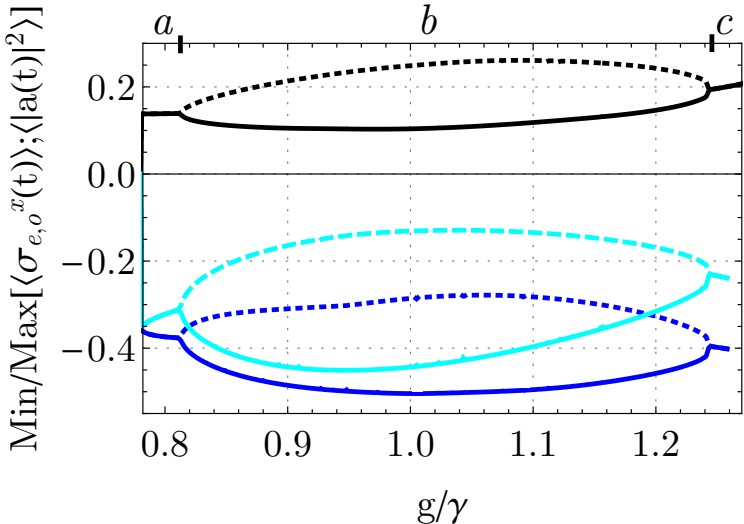

Figure 10: Amplitudes of the oscillations in the (AFM+SR)-OSC phase. Dashed (solid) blue lines show $\max(\min)[\langle \sigma_{e,o}^x(t) \rangle]$ and black lines show $\max(\min)[\langle |a(t)|^2 \rangle]$. Data is obtained by extracting the minimum and maximum of the amplitudes of $\langle \sigma_{e,o}^x(t) \rangle$ and $\langle |a(t)|^2 \rangle$ in a time interval chosen such that it contains several oscillations (if any are present) at long times. If the minimum and maximum coincide, the system settled into a steady state $(a, c)$ corresponding to the (AFM+SR) phase, otherwise the system is in the (AFM+SR)-OSC limit cycle phase $(b)$. Close to the (AFM+SR)-phase, the amplitudes decay continuously. Parameters: $(\omega_0/\gamma = 2.0, \Delta/\gamma = 0.15, \kappa/\gamma = 0.2, V/\gamma = 1.8)$

where the coupling constant $J(\kappa, \omega_0)$ is given by Eq. (53). Here the $\pm$ sign indicates that there are two branches of which only one is stable. Stable solutions ($SR_{UNI}$) are depicted in Fig. 4a. In the limit of weak spin relaxation, it can be seen from Eq. (44) that a stable solution to zeroth-order in $\gamma$ must satisfy $V > \Delta/2$. The phase boundary between the two oscillating phases depicted in Fig. 4 is obtained by comparing the oscillation amplitudes (in Fig. 4b) on the even vs. the odd sublattices in the long-time limit that result from direct integration of Eqs. (12-15).

Next, we turn our attention to the features of the phase diagram depicted in Fig. 5 where losses in the atomic channel can be strong. We analyze the oscillations of both atomic and photonic components in the long-time limit by explicit integration of Eqs. (12-15). Numerically we find persistent oscillations close to the (AFM+SR) region that are different on the even/odd sublattice, see Fig. 6. We determine the phase boundary for the (AFM+SR)-OSC phase depicted in Fig. 5 by sampling initial conditions for the atomic components on the Bloch sphere and then integrating the set of Eqs. (12-15) directly. The phase boundary is set by the parameters $(\Delta/\gamma, g/\gamma)$ for which the long-time limit is determined by the empty atom-cavity system ($FP_{\downarrow}$) for all initial conditions.

In Fig. 10 we track the behaviour of the amplitude of the oscillations as a function of $g/\gamma$ and observe that the amplitudes decay continuously as the (AMR+SR) phase is approached. Numerically, we find no evidence that the (AFM+SR)-phase becomes unstable towards Hopf bifurcations meaning that stable limit cycles occur only outside the AFM+SR phase.

We continue our analysis by describing the behaviour of the magnetisations in the different domains of the phase diagram depicted in Fig. 5. In Fig. 11a we plot the magnetisation values in the steady-state for increasing the atom-light coupling $g/\gamma$. Starting in the empty atom-cavity, the system changes discontinuously into a $SR_{UNI}$ phase that soon after becomes unstable towards even/odd sublattice magnetisations (AFM+SR) that disappear again in favour for a re-

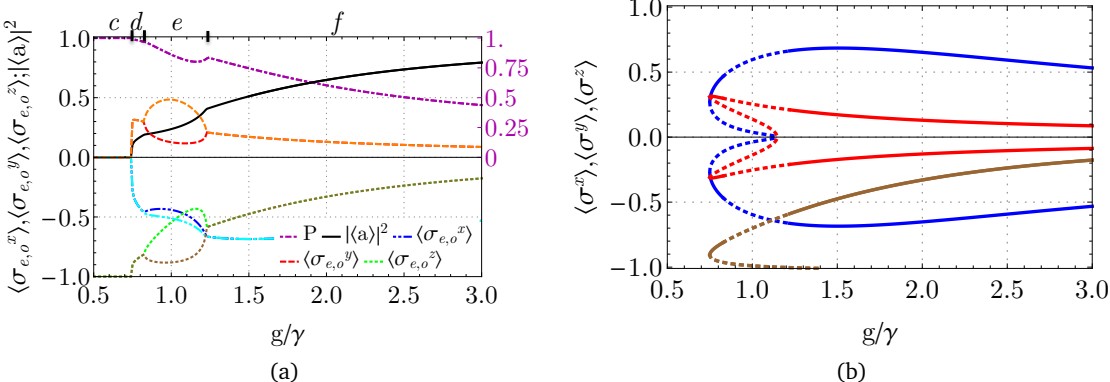

Figure 11: Non-monotonous behaviour of the order-parameters as the atom-light coupling $g/\gamma$ is varied. The system changes discontinuously from the empty atom-cavity system ($\text{FP}_\downarrow$,$c$) into a homogeneous phase ($\text{SR}_{\text{UNI}}$,$d$) that becomes unstable towards an (AFM+SR,$e$) phase that disappears again in favor for an ($\text{SR}_{\text{UNI}}$,$f$) phase. On the right axis, the purity is shown that consistently decays, indicating the transition into a mixed state. (b) Stability analysis for homogeneous solutions as plotted in (a). (Unstable) stable, homogeneous solutions are plotted as (dotted) thick lines. The transition from the $\text{FP}_\downarrow$ state into the $\text{SR}_{\text{UNI}}$ state is discontinuous if $V/\gamma > 0$. Parameters: $(\omega_0/\gamma = 2.0, \Delta/\gamma = -0.1, \kappa/\gamma = 0.2, V/\gamma = 1.8)$

entrance of the $\text{SR}_{\text{UNI}}$ phase. We depict the homogeneous solutions $\text{SR}_{\text{UNI}}$ and their stability in Fig. 11b. We find that for $V/\gamma > 0$ the transition into the Dicke superradiance state is discontinuous.

With $\gamma \neq 0$, the length of the semi-classical Bloch vector $\langle \mathbf{S}_{e,o} \rangle = \left( \langle \sigma_{e,o}^x \rangle, \langle \sigma_{e,o}^y \rangle, \langle \sigma_{e,o}^z \rangle \right)$ is not conserved any more and can shrink for increasing $g/\gamma$ values. In equilibrium systems, an increase in the coupling parameter should stabilize the order in the steady-state, here we instead observe a non-monotonous behaviour where the 'order parameters' decay again after having reached a maximum value. We illustrate this decay by plotting the purity $P = \text{Tr}[(\rho_e \otimes \rho_o)^2] = \text{Tr}[\rho_e^2]\text{Tr}[\rho_o^2]$ of the density matrix alongside the magnetisations. Both quantities decay as $g/\gamma$ is increased. In the case of the purity $P$ this indicates the decay towards a purely mixed state.

The phase transitions in and out of the (AFM+SR) phases are continuous, whereas transitions from the empty atom-cavity system into the $\text{SR}_{\text{UNI}}$ phase are discontinuous for $V/\gamma > 0$, see Fig. 11. On a mean-field level we observe bistabilities in the phase diagram depicted in Fig. 5. These could be induced by the nonlinearities in the mean-field master equations or can hint at non-trivial behaviour induced by fluctuations where the system in the long-time limit switches between the two steady-states predicted on a mean-field level [88]. Mostly, bistabilities occur with the empty atom-cavity system ($\text{FP}_\downarrow$). The corresponding stability line can be calculated analytically from the stability matrix and we find that it is independent of $V$ since the Rydberg-dressed interaction is conditioned on population in the excited state, see Eq. (42). The size of the (AFM+SR) region instead does depend on $V$.

## 4.4 Discussion of beyond mean-field effects

Our analysis is based on mean-field theory and in this section we briefly discuss effects not captured by our approach and alternative theoretical approaches used in the literature. One possibility to capture fluctuations in far-from-equilibrium quantum spin systems is the real-time quantum field theory approach [89, 90] representing the spins as (Majorana) fermions.

The number of fermion "flavors" and possibly constraints/gauge fields make this a complex endeavour, but, once developed, will be capable of treating even very large systems.

Several studies investigated effects beyond mean field in driven-dissipative lattice models that allow to acquire some intuition for the effect of correlations and fluctuations and for the validity of single-site mean-field studies in driven-lattice models out of equilibrium. Specifically, various numerical techniques such as variational approaches [91,92], cluster models [87,93], matrix product states [39], or quantum trajectories [88,94] can help to shed light on the effects of correlations and fluctuations in the steady-state. However, most of these studies have focussed on a single short-range interaction. Using these techniques, it was, for example, found that bistable regions may become washed out, when some form of correlation is taken into account [91,92,95].

The most dramatic modifications of the mean-field behaviour can be expected in one-dimensional systems. First, the loss processes will lead to a noise-induced effective temperature for the atomic spins [53,63] and modify the distribution function of the atoms; to capture the latter effect a quantum kinetic analysis is necessary [59]. In the presence of an effective temperature, it is known since Polyakov [96] that instantons prohibit one-dimensional Ising systems from ordering on longest scales. This implies, that we expect that the AFM phases in Fig. 2 will disappear for a 1D spin array (whether they will be replaced by SR phases or the vacuum depends on the sign of the detunings). As in 1d quantum systems, also here despite the absence of true long-range order, even-odd correlations will be visible in the correlation functions and structure factors, whose computation, however, can get quite involved/is not known in the absence of equilibrium. The order parameter of the SR phases will of course remain stable, due to the effectively infinite range of the interaction, dimensionality is irrelevant here.

Our analysis was, however, focusing on a two-dimensional setup where not only SR but also the AFM order can be stable in the presence of thermal and non-equilibrium fluctuations as domain walls cost an infinite amount of energy. For the two-dimensional spin array discussed in the present paper, we believe the qualitative features are robust, i.e. fluctuations might shift the phase transitions, induce a finite effective temperature, and further broaden the spectra but will not fully destroy the order. In particular the AFM+SR strip ending in the multi-critical point shown in Fig. 2 might get washed out and/or replaced by a first order transition. Mean-field also does not capture all critical properties. Note that not much is known about first-order transitions in multi-critical, driven-dissipative systems. Perhaps the most difficult question is whether the oscillating phases shown in the phasediagram of Fig. 5 will survive fluctuations beyond mean field. Here it is important to note that due to the infinite range interaction mediated by the cavity, a spontaneous breaking of time-translational symmetry is possible in two (and even one) dimension and is not destroyed by goldstone-mode fluctuations [27]. Nevertheless, it is unclear whether the heating processes naturally occuring in periodically driven interacting many-particle systems [97] will be compensated by cooling processes due to radiative losses in such a way that the oscillations survive.

# 5   Conclusions and future directions

The point of this paper was to create a base case (model and approximate solution) for a large array of *self-interacting atomic qubits* coupled to a single-mode optical light field. Why do we believe this is needed? Because there is an increasing number of experimental platforms ranging from ultracold atoms in optical cavities, superconducting circuits, photonic pulses travelling through Rydberg gases to nano-photonic crystals, seeking to scale up the number of qubits and interface them with photons. Qubit-qubit interactions can be wanted –to me-

diate photon-photon nonlinearities for example– or stray, in which case they would be seen to dephase a collective coupling of a set of qubits to photons. In the face of the considerable complexity these systems generate –large number of quantum spins, fluctuating light fields, non-equilibrium aspects– our simple model has yielded some experimentally directly testable predictions: A regime where magnetic translation symmetry breaking and superradiance occur together, a new even-odd collective mode in the cavity spectrum, and intriguing, oscillating solutions for both, the spin components and a coherent photon field. Present-day technology with Rydberg-dressed spin lattices in optical cavities should be able to check and refine these results. Unfortunately, we were not able to solve even our simple model exactly; the Rydberg-mediated nearest-neighbour interaction does induce non-trivial quantum fluctuations (centered around the even-odd modulation momentum $(\pi, \pi)$) between the spins. Our non-equilibrium mean-field ansatz for the density matrix kept track of only the expectation values of the spin components on the even and odd sub-lattices, and the photon field, respectively. We are not aware of a developed technique (see Sec. 4.4 for a brief discussion), which can capture quantum fluctuations for large, far-from-equilibrium quantum spin systems coupled to the (potentially large) Hilbert space of one or multiple photon modes. Promising efforts in this direction invoke a fermion representation of the quantum spins on the closed-time Keldysh contour ([89, 90] and references therein); this then, in principle, allows to leverage over diagrammatic techniques well-developed for ground state fermions. Work in this direction is underway. Particularly promising physical set-ups to study the interplay of interacting qubits with light in the future are nano-photonic and one-dimensional quantum-optical systems [34, 35, 98–103], in which huge effects from even small qubit-qubit interactions can be expected. Moreover, these systems typically contain an (infinite) continuum of photon modes significantly enriching the complexity of the photonic Hilbert space at one's disposal. The same is true for optical resonators in multi-mode operation [29, 104, 105] making them also interesting targets for further explorations.

## Acknowledgments

We thank A. Glätzle, C. Kollath, and P. Zoller for good discussions and further acknowledge helpful remarks by T. Pohl, P. Rabl, H. Ritsch, and J. Zeiher during the Quantum Optics 2016 conference. JG thanks the Department of Physics at Harvard University for hospitality, where this research was partly carried out (during the first half of 2015). We are grateful to M. D. Lukin for discussions on related topics. This work was supported by the Leibniz Prize of A. Rosch, by the Harvard-MIT Center for Ultracold Atoms (CUA), by the Multidisciplinary University Research Initiative (MURI), and by the German Research Foundation (DFG) through CRC 1238 and the Institutional Strategy of the University of Cologne within the German Excellence Initiative (ZUK 81).

## A  Mean-field solution of the T=0 equilibrium spin model

In this section, we analyze Hamiltonians $H_{\text{spin−light}}$ Eq. (2) and $H_{\text{spin}}$ Eq. (1) within a (standard) equilibrium mean-field theory for spins. We will find similar phases to those in Fig. 2 upon identifying one of the spin-spin interaction constants with cavity parameters, somewhat surprisingly including the photon decay $\kappa$. The deeper reason for this is that with the counter-rotating terms, in the atom-light coupling the excitation number is stabilized despite the loss rates. Accordingly, the non-equilibrium steady-state phase in the long-time limit are then qualitatively similar to the ground state phases. Dynamics and statistics (effective temperature) of

the full non-equilibrium system, however, remain qualitatively drastically different.

First, we integrate out the quadratic photon terms which yields an effective Hamiltonian that features a ferromagnetic all-to-all atom-atom coupling $J$. The connection to the non-equilibrium system is then made explicit by letting $J$ depend on $\kappa$ as pointed out below. The Hamiltonian we consider is written as

$$\tilde{H} = -\frac{J}{N} \sum_{ij} \sigma_i^x \sigma_j^x - \frac{\Delta}{2} \sum_i \sigma_i^z + \frac{1}{2} V \sum_{\langle \ell m \rangle}^N \sigma_\ell^{ee} \sigma_m^{ee} \tag{45}$$

Where the $1/2$ in front of the Rydberg interaction avoids overcounting. We cast the last terms into a spin-language with the replacement $\sigma_\ell^{ee} = 1/2(1 + \sigma_\ell^z)$. We decouple the interaction terms in mean-field theory by expanding the operators around their mean-value to linear order in fluctuations. We neglect all second-order fluctuation terms and write the effective spin-Hamiltonian in a form that resembles the interaction of the spin-variables with an effective, local magnetic field that needs to be determined self-consistently and represents the mean field from the neighbouring spins. Ignoring constant energy shifts, the full mean-field Hamiltonian assuming $d = 2$-spatial dimensions is given as

$$\tilde{H}^{\mathrm{MF}} = -V\frac{N}{2} \langle \sigma_{\mathrm{even}}^z \rangle \langle \sigma_{\mathrm{odd}}^z \rangle + NJ \langle \sigma^x \rangle^2 + \sum_{i \in even} \boldsymbol{B}^{\mathrm{even}} \boldsymbol{\sigma}_i + \sum_{j \in odd} \boldsymbol{B}^{\mathrm{odd}} \boldsymbol{\sigma}_j \tag{46}$$

Here, we have already accounted for an $even/odd$ sub-lattice asymmetry in the $z-$components. We use the vector of Pauli matrices as $\boldsymbol{\sigma} = (\sigma^x, \sigma^y, \sigma^z)^T$ and define the local magnetic fields as

$$\boldsymbol{B}^{\mathrm{even/odd}} = \left( \left[ \langle \sigma_{\mathrm{odd/even}}^z \rangle V + (V - \Delta/2) \right] \hat{\boldsymbol{z}} + 2J \langle \sigma^x \rangle \hat{\boldsymbol{x}} \right) \tag{47}$$

We evaluate the partition sum

$$Z = \mathrm{Tr}\left[ e^{-\beta \tilde{H}^{\mathrm{MF}}} \right] = 2^N \left[ \cosh(|\boldsymbol{B}^{\mathrm{even}}|) \cosh(|\boldsymbol{B}^{\mathrm{odd}}|) \right]^{N/2} \exp\left[ V\frac{N}{2} \beta \langle \sigma_{\mathrm{even}}^z \rangle \langle \sigma_{\mathrm{odd}}^z \rangle - N\beta J \langle \sigma^x \rangle^2 \right] \tag{48}$$

to obtain the self-consistency equations for the order-parameters

$$\phi = \frac{\langle \sigma_{\mathrm{even}}^x \rangle + \langle \sigma_{\mathrm{odd}}^x \rangle}{2} = \frac{1}{2} \tanh\left(\beta|\boldsymbol{B}^{\mathrm{odd}}|\right) \frac{B_x^{\mathrm{odd}}}{|\boldsymbol{B}^{\mathrm{odd}}|} + \frac{1}{2} \tanh\left(\beta|\boldsymbol{B}^{\mathrm{even}}|\right) \frac{B_x^{\mathrm{even}}}{|\boldsymbol{B}^{\mathrm{even}}|} \tag{49}$$

$$\rho = \frac{\langle \sigma_{\mathrm{even}}^z \rangle - \langle \sigma_{\mathrm{odd}}^z \rangle}{2} = \frac{1}{2} \tanh(\beta|\boldsymbol{B}^{\mathrm{even}}|) \frac{B_z^{\mathrm{even}}}{|\boldsymbol{B}^{\mathrm{even}}|} - \frac{1}{2} \tanh\left(\beta|\boldsymbol{B}^{\mathrm{odd}}|\right) \frac{B_z^{\mathrm{odd}}}{|\boldsymbol{B}^{\mathrm{odd}}|} \tag{50}$$

$$\rho_0 = \frac{\langle \sigma_{\mathrm{even}}^z \rangle + \langle \sigma_{\mathrm{odd}}^z \rangle}{2} = \frac{1}{2} \tanh(\beta|\boldsymbol{B}^{\mathrm{even}}|) \frac{B_z^{\mathrm{even}}}{|\boldsymbol{B}^{\mathrm{even}}|} + \frac{1}{2} \tanh\left(\beta|\boldsymbol{B}^{\mathrm{odd}}|\right) \frac{B_z^{\mathrm{odd}}}{|\boldsymbol{B}^{\mathrm{odd}}|} \tag{51}$$

Where $\rho$ is the staggered magnetisation and $\rho_0$ is the average magnetisation in the $z$ direction. The magnetic order parameter $\phi$ measures the magnetisation in $x$-direction and $\beta = 1/T$ is the inverse temperature. We denote the free energy per spin in the zero temperature limit $T \to 0$ as

$$f = \left. \frac{F}{N} \right|_{T \to 0} = \left. -\frac{T}{N} \log(Z) \right|_{T \to 0} = \frac{1}{2} V \left( \rho^2 - \rho_0^2 \right) + J\phi^2$$

$$- \frac{1}{2} \left( \sqrt{\left( V(\rho_0 - \rho) + V - \frac{\Delta}{2} \right)^2 + 4J^2\phi^2} + \sqrt{\left( V(\rho + \rho_0) + V - \frac{\Delta}{2} \right)^2 + 4J^2\phi^2} \right) \tag{52}$$

We can determine the zero-temperature phase-diagram by solving the coupled set of equations (49 − 51) numerically and retain only the solutions with the lowest free-energy according to Eq. (52). We obtain the splitting in the $(\langle \sigma^x_{\text{even}} \rangle, \langle \sigma^x_{\text{odd}} \rangle)$ components by using Eq. (16). We find that we can map the equilibrium phase-diagram to the phase diagram obtained by calculating the non-equilibrium steady-states (see Fig. 2) if we identify the ferromagnetic exchange coupling as

$$J(g, \kappa) = \frac{g^2 \omega_0}{\omega_0^2 + \kappa^2} \tag{53}$$

This coupling is inferred from solving for the steady-state values of the photons (see Eq. (15)) which is given as $g\left(\langle a \rangle + \langle a^\dagger \rangle\right) = -\frac{1}{2} g \langle \sigma^x_{\text{even}} + \sigma^x_{\text{odd}} \rangle \left( \frac{g}{\omega - i\kappa} + \frac{g}{\omega + i\kappa} \right) \propto J(g, \kappa)$. Allowing the interpretation that the photonic losses with rate $\kappa$ weaken the atom-atom couplings.

## B  Transformation of fully time-dependent model into rotating frame

Here, we detail calculations where we derive how the parameters of the Hamiltonians $H_{\text{spin–light}}$ given by Eq. (2) and $H_{\text{spin}}$ given by Eq. (1) are related to tunable laser parameters that result from the optical implementation shown in Fig. 7. The Hamiltonian we consider is of the form

$$H = H_{\text{cav}} + H_{\text{atoms}} + H(t)_{\text{pump}} + H_{\text{atom–light}} + H_{\text{atom–atom}} \tag{54}$$

$$H_{\text{cav}} = \omega_0 a^\dagger a \tag{55}$$

$$H_{\text{atoms}} = \sum_{\ell=1}^{N} \omega_d |d\rangle_\ell \langle d| + \omega_e |e\rangle_\ell \langle e| + \omega_{\text{Ryd}} |\text{Ryd}\rangle_\ell \langle \text{Ryd}| + \omega_1 |1\rangle_\ell \langle 1| \tag{56}$$

$$H_{\text{pump}}(t) = \sum_{\ell=1}^{N} \frac{\Omega_e}{2} e^{-i\omega_{\Delta e} t} |e\rangle_\ell \langle 1| + \frac{\Omega_d}{2} e^{-i\omega_{\Delta d} t} |d\rangle_\ell \langle 0| + \frac{\Omega_{\text{Ryd}}}{2} e^{-i\omega_{\Delta r} t} |\text{Ryd}\rangle_\ell \langle 1| + \text{h.c.} \tag{57}$$

$$H_{\text{atom–light}} = \sum_{\ell=1}^{N} \left( g_d |d\rangle_\ell \langle 1| + g_e |e\rangle_\ell \langle 0| \right) a + \text{h.c.} \tag{58}$$

$$H_{\text{atom–atom}} = \sum_{\langle \ell m \rangle} V_{\ell m} \left( |\text{Ryd}\rangle_\ell \langle \text{Ryd}| \right) \left( |\text{Ryd}\rangle_m \langle \text{Ryd}| \right) \tag{59}$$

The frequencies $(\omega_d, \omega_e, \omega_{\text{Ryd}}, \omega_1)$ refer to the atomic levels labelled by the sequence $(d, e, \text{Ryd}, 1)$ and are measured relative to the atomic level $|0\rangle$. Correspondingly, the frequencies $(\omega_{\Delta d}, \omega_{\Delta e}, \omega_{\Delta r})$ refer to the laser frequencies of the pump-terms. Here, $\omega_0$ denotes the bare cavity resonance. We have assumed homogeneous pumping of the atoms from the side $\Omega_{(d,e);\ell} \approx \Omega_{(d,e)}$ and a homogeneous coupling of the light field to the atoms $g_{(d,e);\ell} \approx g_{(d,e)}$. We eliminate the explicit time-dependence by switching into a rotating frame such that the new Hamiltonian reads

$$\tilde{H} = U^\dagger H_0 U - U^\dagger i \partial_t U \tag{60}$$

where the Hamiltonian $H_0$ is given as

$$U(t) = \exp(-i H_0 t) \tag{61}$$

$$H_0 = \left( \omega_{\Delta d} - \omega_1' \right) a^\dagger a$$
$$+ \sum_{\ell=1}^{N} \left[ \left( \omega_{\Delta e} + \omega_1' \right) |e\rangle_\ell \langle e| + \omega_{\Delta d} |d\rangle_\ell \langle d| + \omega_1' |1\rangle_\ell \langle 1| + \left( \omega_{\text{Ryd}} + \omega_1' \right) |\text{Ryd}\rangle_\ell \langle \text{Ryd}| \right] \tag{62}$$

Cross coupling lasers need to be tuned such that they are strongly detuned from the levels $(|d\rangle, |e\rangle)$ which can then be eliminated adiabatically. Under the condition $|\Delta_{d,e}| \gg \kappa, \gamma, \Omega_{d,e}$, the dynamics of the system are now described by an effective Hamiltonian $\tilde{H} = \tilde{H}_{\mathrm{Ryd}} + \tilde{H}_{10} + \tilde{H}_L$:

$$\tilde{H}_L = \sum_{\ell=1}^{N} \frac{\Omega_{\mathrm{Ryd}}}{2} \left( |\mathrm{Ryd}\rangle_\ell \langle 1| + |1\rangle_\ell \langle \mathrm{Ryd}| \right) \tag{63}$$

$$\tilde{H}_{\mathrm{Ryd}} = -\Delta_{\mathrm{Ryd}} \sum_{\ell=1}^{N} |\mathrm{Ryd}\rangle_\ell \langle \mathrm{Ryd}| + \sum_{\langle \ell m \rangle} V_{\ell m} \left( |\mathrm{Ryd}\rangle_\ell \langle \mathrm{Ryd}| \right) \left( |\mathrm{Ryd}\rangle_m \langle \mathrm{Ryd}| \right) \tag{64}$$

$$\begin{aligned}
\tilde{H}_{10} = {}& \omega_a a^\dagger a + \sum_{\ell=1}^{N} \Bigg[ \left( \Delta_1 + \frac{\Omega_e^2}{4\Delta_e} \right) |1\rangle_\ell \langle 1| + \frac{\Omega_d}{4\Delta_d} |0\rangle_\ell \langle 0| \\
& + \frac{1}{2} \left( \frac{g_e \Omega_e}{\Delta_e} |0\rangle_\ell \langle 1| a^\dagger + \frac{g_d \Omega_d}{\Delta_d} |1\rangle_\ell \langle 0| a^\dagger + \mathrm{h.c.} \right) + \left( \frac{g_e^2}{\Delta_e} |0\rangle_\ell \langle 0| + \frac{g_d^2}{\Delta_d} |1\rangle_\ell \langle 1| \right) a^\dagger a \Bigg]
\end{aligned} \tag{65}$$

where we have used the following frequencies

$$\begin{aligned}
\omega_a &= \omega_0 - \left( \omega_{\Delta d} - \omega_1' \right), \\
\Delta_{\mathrm{Ryd}} &= -\left[ \omega_{\mathrm{Ryd}} - \left( \omega_{\Delta r} + \omega_1' \right) \right] \\
\Delta_1 &= \omega_1 - \omega_1', \\
2\omega_1' &= \omega_{\Delta d} - \omega_{\Delta e}.
\end{aligned} \tag{66}$$

In a next step, high-lying Rydberg states are admixed to the ground-states $|1\rangle_\ell$ to realise a Rydberg-dressed interaction between the states $|\tilde{1}\rangle = |1\rangle + \frac{\Omega_{\mathrm{Ryd}}}{2\Delta_{\mathrm{Ryd}}} |\mathrm{Ryd}\rangle + \mathcal{O}\left( \frac{\Omega_{\mathrm{Ryd}}}{2\Delta_{\mathrm{Ryd}}} \right)$. Typically, two-body Born-Oppenheimer potentials as a function of the distance $r_{ij}$ between two Rydberg levels are obtained by diagonalising Hamiltonians of the form $H_L + H_{\mathrm{Ryd}}$ in a two-atom basis [72]. A detailed calculation that includes coupling to the complicated level structure is thus highly non-trivial. Focusing on the weak-dressing regime $\Omega_{\mathrm{Ryd}}/\Delta_{\mathrm{Ryd}} \ll 1$ and red-detuning of the dressing laser we follow the many-body perturbation expansion performed in Ref. [71] to obtain the effective Hamiltonian for the Rydberg part to leading order in the corrections

$$\begin{aligned}
\tilde{H}_{\mathrm{Ryd}} &= -\frac{\Omega_{\mathrm{Ryd}}^2}{4\Delta_{\mathrm{Ryd}}} \sum_{\ell=1}^{N} |\tilde{1}\rangle_\ell \langle \tilde{1}| + \frac{1}{2} \left( \frac{\Omega_{\mathrm{Ryd}}}{2\Delta_{\mathrm{Ryd}}} \right)^4 \sum_{i \neq j} \frac{C_6}{r_{ij} + R_c^6} \left( |\tilde{1}\rangle_i \langle \tilde{1}| \right) \left( |\tilde{1}\rangle_j \langle \tilde{1}| \right) \\
&= -\frac{\Omega_{\mathrm{Ryd}}^2}{4\Delta_{\mathrm{Ryd}}} \sum_{\ell=1}^{N} |\tilde{1}\rangle_\ell \langle \tilde{1}| + \frac{1}{2} \sum_{i \neq j} V_{ij}^{\mathrm{eff}} \left( |\tilde{1}\rangle_i \langle \tilde{1}| \right) \left( |\tilde{1}\rangle_j \langle \tilde{1}| \right)
\end{aligned} \tag{67}$$

It can be seen that the dressed states $|\tilde{1}\rangle$ acquire additional light-shifts $\sim \Omega_{\mathrm{Ryd}}^2/(4\Delta_{\mathrm{Ryd}})$ and the Rydberg potential is tunable by changing $(\Omega_{\mathrm{Ryd}}, \Delta_{\mathrm{Ryd}})$. Here, $V_{ij}^{\mathrm{eff}}$ and $R_c$ are defined in Eq. (10). We now replace $|1\rangle$ with the dressed Rydberg state $|\tilde{1}\rangle$ everywhere in Eq. (65). With

$$\sum_{\ell=1}^{N} |1\rangle_\ell \langle 1| = \frac{1}{2} \sum_{\ell=1}^{N} \left( |1\rangle_\ell \langle 1| - |0\rangle_\ell \langle 0| + N \right) = \frac{1}{2} \sum_{\ell=1}^{N} \sigma_\ell^z + \frac{N}{2} \tag{68}$$

$$\sum_{\ell=1}^{N} |0\rangle_\ell \langle 0| = -\frac{1}{2} \sum_{\ell=1}^{N} \left( |1\rangle_\ell \langle 1| - |0\rangle_\ell \langle 0| - N \right) = -\frac{1}{2} \sum_{\ell=1}^{N} \sigma_\ell^z + \frac{N}{2} \tag{69}$$

the Hamiltonian is now cast into the form:

$$
\begin{aligned}
\tilde{H} =& a^\dagger a \left[ \frac{N}{2} \left( \frac{g_e^2}{\Delta_e} + \frac{g_d^2}{\Delta_d} \right) + \omega_a \right] \\
&+ \sum_{\ell=1}^{N} \sigma_\ell^z \frac{1}{2} \left[ \left( \frac{\Omega_d^2}{4\Delta_d} - \frac{\Omega_e^2}{4\Delta_e} \right) + \Delta_1 - \frac{\Omega_{\text{Ryd}}^2}{4\Delta_{\text{Ryd}}} + \frac{1}{2} \sum_{m(m\neq\ell)}^{N} V_{m\ell}^{\text{eff}} \right] + \sum_{\ell=1}^{N} \sigma_\ell^z \frac{1}{2} \frac{1}{N} \left( \frac{g_d^2}{\Delta_d} - \frac{g_e^2}{\Delta_e} \right) a^\dagger a \\
&+ \sum_{\ell=1}^{N} \left[ \frac{\lambda_d}{\sqrt{N}} \left( \sigma_\ell^+ a^\dagger + \sigma_\ell^- a \right) + \frac{\lambda_e}{\sqrt{N}} \left( \sigma_\ell^+ a + \sigma_\ell^- a^\dagger \right) \right] + \frac{1}{2} \sum_{i\neq j} V_{\text{ij}}^{\text{eff}} \frac{\sigma_i^z}{2} \frac{\sigma_j^z}{2} ,
\end{aligned}
\tag{70}
$$

with effective spin-photon couplings (set equal and denoted by $g$ in Eq. (9))

$$
\lambda_{d,e} = \sqrt{N} \frac{g_{d,e} \Omega_{d,e}}{2\Delta_{d,e}} .
\tag{71}
$$

To generate AFM ordering it is advantageous for the effective longitudinal field corresponding to the second term in the first line in Eq. (70) to be negative. We analyse typical orders of magnitudes. The hyperfine structure splitting in the ground state manifold is $\omega_1 = 2\pi \times 6.835\text{GHz}$. Typically the cavity-assisted Raman transitions are achieved by coupling to the first excited state manifold that is split into a fine-structure $5^2 P_{1/2}$ with $F' = 2$ and $F' = 1$ that for this choice is on the order of 812MHz. The external driving lasers are separated by approximately twice the ground-state hyperfine splitting $\omega_{1'} = \frac{1}{2} (\omega_{\Delta d} - \omega_{\Delta e}) \sim \omega_1$ such that $\Delta_1 = \omega_1 - \omega_{1'} \sim$MHz. For weakly admixing the Rydberg state to the groundstate manifold $|\uparrow\rangle$ the detuning from the Rydberg state $\Delta_{\text{Ryd}}$ must satisfy $\Omega_{\text{Ryd}} \ll \Delta_{\text{Ryd}}$. Typical Rabi frequencies for the drive to the Rydberg level are $\Omega_{\text{Ryd}} \sim$MHz. The detuning from the Rydberg level now has a frequency component $\omega_{1'}$ from the Raman-scheme: $\Delta_{\text{Ryd}} = -[\omega_r - (\omega_{\Delta r} + \omega_{1'})]$. This can take the usual detuning $\Delta_{\text{Ryd}}$ far above the MHz regime which makes $\Delta_{\text{Ryd}} \ggg \Omega_{\text{Ryd}}$. The longitudinal field $\left( \Delta_1 - \frac{\Omega_{\text{Ryd}}^2}{4\Delta_{\text{Ryd}}} \right)$ for $\Omega_d = \Omega_e$ and $\Delta_d = \Delta_e$ is in the MHz range and can in principle be tuned positive and negative.

## C Validity analysis of the even-odd sublattice Ansatz

Here, we determine the linear stability of the homogeneous fixed points of Eqs. (72-76) against excitations with momentum $\boldsymbol{k}$, see e.g. References [42, 44]. In driven-dissipative lattice models with short-range interaction, orderings with incommensurate wavevectors have been observed [41–45]. This can happen because of an interplay of dissipation and a competition of different, momentum-dependent interactions such as in a driven spin-1/2 XYZ-model [27]. In driven-dissipative Bose-Hubbard-type lattice models (see e.g. Reference [43]) multimode photon fields are considered which have a finite momentum dependence. This is in contrast to our single-mode photon field that only couples to the zero-momentum component. The infinite-range atom-light and the antiferromagnetic spin-spin interaction suggest that the steady-states can either be uniform or that it can break the translational invariance of the system, respectively (see also Table 1). We thus expect and find that homogeneous mean field solutions are maximally unstable either at $(k_x, k_y) = (0, 0)$ or against excitations with $(k_x, k_y) = (\pi, \pi)$. We outline the analysis below.

On a mean-field level, we write down the master-equation for every lattice site $\boldsymbol{n}$

$$\partial_t \langle \sigma_n^x(t) \rangle = \langle \sigma_n^y(t) \rangle [\Delta - \frac{V}{2} \sum_{\langle nm \rangle} (\langle \sigma_m^z(t) \rangle + 1)] - \frac{\gamma}{2} \langle \sigma_n^x(t) \rangle \tag{72}$$

$$\partial_t \langle \sigma_n^y(t) \rangle = \langle \sigma_n^x(t) \rangle [\frac{V}{2} \sum_{\langle nm \rangle} (\langle \sigma_m^z(t) \rangle + 1) - \Delta]$$

$$- 2g[\langle a(t) \rangle + \langle a^\dagger(t) \rangle] \langle \sigma_n^z(t) \rangle - \frac{\gamma}{2} \langle \sigma_n^y(t) \rangle \tag{73}$$

$$\partial_t \langle \sigma_n^z(t) \rangle = 2g[\langle a(t) \rangle + \langle a^\dagger(t) \rangle] \langle \sigma_n^y(t) \rangle - \gamma(1 + \langle \sigma_n^z(t) \rangle) \tag{74}$$

$$\partial_t \langle a(t) \rangle = -(\kappa + i\omega_0) \langle a(t) \rangle - ig \sum_n \langle \sigma_n^x(t) \rangle \tag{75}$$

$$\partial_t \langle a^\dagger(t) \rangle = -(\kappa - i\omega_0) \langle a(t) \rangle + ig \sum_n \langle \sigma_n^x(t) \rangle, \tag{76}$$

where $\boldsymbol{n}$ is a two-dimensional position vector on the square lattice. We check the validity of our even-odd sublattice approach by adding plane-wave perturbations to the uniform steady-states with the Ansatz

$$\langle \boldsymbol{\sigma_n}(t) \rangle = \langle \boldsymbol{\sigma} \rangle + \delta \boldsymbol{\sigma_n}(t), \quad \langle a(t) \rangle = \langle a \rangle + \delta a(t), \tag{77}$$

where $\boldsymbol{\sigma} = (\langle \sigma^x \rangle, \langle \sigma^y \rangle, \langle \sigma^z \rangle)^T$ are the homogeneous solutions to Eqs. (72-76) and $\boldsymbol{k}$ contains the wave numbers of the perturbation. We Fourier transform according to

$$\delta \boldsymbol{\sigma_n}(t) = \frac{1}{N} \sum_{\boldsymbol{k}} e^{i\boldsymbol{k} \cdot \boldsymbol{n}} \delta \boldsymbol{\sigma_k}(t), \quad \boldsymbol{k} = (k_x, k_y)^T, \quad k_\ell = \frac{2\pi}{N} j, \quad j = 0, \dots, N-1 \tag{78}$$

We linearize equations Eqs. (72-76) in the fluctuations $(\delta \boldsymbol{\sigma_k}(t), \delta a(t))$ and obtain a set of equations for each wave-vector $\boldsymbol{k}$

$$\partial_t \boldsymbol{\delta_k}(t) = \mathscr{D}_{\boldsymbol{k}} \boldsymbol{\delta_k}(t) \tag{79}$$

$$\boldsymbol{\delta_k}(t) = \left( \delta \sigma_k^x(t), \delta \sigma_k^y(t), \delta \sigma_k^z(t), \delta a(t), \delta a^\dagger(t) \right)^T \tag{80}$$

with the stability matrix

$$\mathscr{D}_{\boldsymbol{k}} = \begin{pmatrix} -\frac{\gamma}{2} & \Delta - 2V(\langle \sigma^z \rangle + 1) & -V \langle \sigma^y \rangle t_{\boldsymbol{k}} & 0 & 0 \\ 2V(\langle \sigma^z \rangle + 1) - \Delta & -\frac{\gamma}{2} & V \langle \sigma^x \rangle t_{\boldsymbol{k}} - 2g(\langle a \rangle + \langle a^\dagger \rangle) & -2g \langle \sigma^z \rangle & -2g \langle \sigma^z \rangle \\ 0 & 2g(\langle a \rangle + \langle a^\dagger \rangle) & -\gamma & 2g \langle \sigma^y \rangle & 2g \langle \sigma^y \rangle \\ -ig\delta(\boldsymbol{k}) & 0 & 0 & -(\kappa + i\omega_0) & 0 \\ +ig\delta(\boldsymbol{k}) & 0 & 0 & 0 & -(\kappa - i\omega_0) \end{pmatrix} \tag{81}$$

here the momentum dependence is given by $t_{\boldsymbol{k}} = \cos(k_x) + \cos(k_y)$. The stability matrix has eigenvalues $\lambda$ that depend on the wave number $\boldsymbol{k}$. The sign ($\pm$) of the real part of the eigenvalues determine if perturbations with momentum $\boldsymbol{k}$ decay (-) or grow (+) in time. If an eigenvalue acquires a positive real part, the uniform solution is unstable. The dynamics of the instability will be dominated by the wave vector $\boldsymbol{k}$ for which Re[$\lambda$] is at its maximum. Inspecting the matrix in Eq. (81), one can see that for an infinite system size, it depends continuously on the momentum $\boldsymbol{k}$ only through the Rydberg interaction which, on a mean-field level, favors ordering around $(k_x, k_y) = (\pi, \pi)$. The appearance of the delta function $\delta(\boldsymbol{k})$ shows that fluctuations in the coherent photon field only couple to uniform perturbations. In particular, there is no competition with other $\boldsymbol{k}$-dependent terms that could induce instabilities at finite momentum $\boldsymbol{k} \neq (0,0)$ and $\boldsymbol{k} \neq (\pi, \pi)$. In Fig. 12 we show where the homogeneous

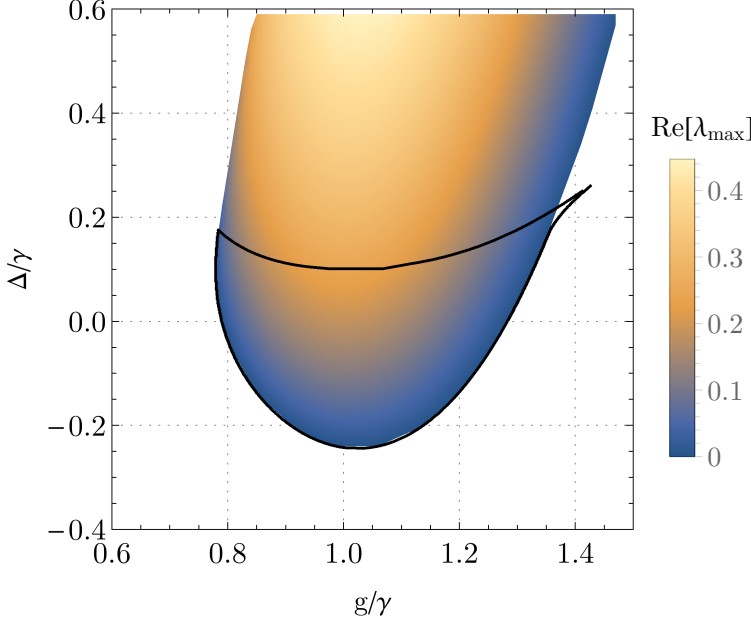

Figure 12: Instability of the homogeneous solution against excitations with wavevector $(k_x, k_y) = (\pi, \pi)$, calculated from Eq. (81). The color scale shows the real part of the eigenvalue that is maximally unstable. Using the even-odd sublattice Ansatz we find the phase-boundary of stable anti ferromagnetic solutions (enclosed by the bold line) which are consistent with the stability analysis of the homogenous solutions. In the upper half-plane, where $\Delta > 0$, there is a region where the homogeneous solution is unstable against excitations with $(k_x, k_y) = (\pi, \pi)$ but the mean-field antiferromagnetic solutions are not stable above the bold line. The region around $(g/\gamma, \Delta/\gamma) \approx (1.4, 0.18)$ is bistable and can show steady-states of $(\text{AFM} + \text{SR})$ or $\text{SR}_{\text{UNI}}$ ordering. This plot is done for the same parameters as in the entire phase diagram that is given in Fig. 5

solution to Eqs. (72-76) (excluding the empty cavity, where $\langle \sigma^x \rangle = \langle \sigma \rangle^y = 0$ and $\langle \sigma^z \rangle = -1$ and $\langle a \rangle = 0$) in linear response is maximally unstable towards excitations at $(k_x, k_y) = (\pi, \pi)$. Within the even-odd sublattice Ansatz of Eqs. (12-15) we include the phase boundary of the stable $(\text{AFM} + \text{SR})$ solutions (solid line). Our results are fully consistent with each other. As it can be seen in Fig. 12, there is a region where the homogeneous solution is unstable towards excitation at $\mathbf{k} = (\pi, \pi)^T$ but where the corresponding antiferromagnetic solution is not a stable steady-state. The entire phase-diagram is given in Fig. 5.

## D    Hierarchy of energy scales and problematic Rydberg decays

We now compare typical timescales associated to the Hamiltonian and Liouvillian dynamics given by Eqs. (1)-(4), using two recently performed experiments. One on a 2d Ising Hamiltonian with an interaction between Rydberg-dressed ground states, (see Eq. (1)) carried out by Zeiher *et al.* [5] and an experiment by Baden *et al.* [52] with cavity-assisted Raman processes to realise the Dicke superradiance transition with ultracold atoms coupled to a high-finesse optical cavity, as described with the Hamiltonian given by Eq. (2). The list of time and frequency scales is given in Table 3 and in Table 2, respectively. It can be seen that the Rydberg-dressed interaction $V$ is relatively small compared to the other appearing energy scales. For an experimental realisation of a phase with an even/odd asymmetry it would thus be required to

| | $\gamma_{\text{BB}}\beta^2/2\pi$ | $\gamma_r\beta^2/2\pi$ | $V/2\pi$ | $\Delta^Z/2\pi$ | $\kappa/2\pi$ | $\Delta^B/2\pi$ | $g_c/2\pi$ | $\omega_0/2\pi$ |
|---|---|---|---|---|---|---|---|---|
| kHz | 0.003-0.020 | 0.06-0.45 | 0.1-1.8 | 27-64 | 100 | 50-100 | 50-150 | 100-300 |

Table 2: Hierarchy of frequencies for all involved energy scales. The energy scales involving the spin-spin dynamics ($\gamma_{\text{BB}}, \gamma_r, V, \Delta^Z$) are calculated from experiments by Zeiher *et al.* [5]. The energy scales ($\kappa, \Delta^B, g_c, \omega_0$) involving the spin-light and cavity dynamics are calculated from the experiments performed by Baden *et al.* [52]. Here, $\Delta^Z$ and $\Delta^B$ refer to the level splitting of the two-level atom and $\omega_0$ is the effective cavity detuning. $g_c$ refers to the critical atom-light coupling for the superradiance transition in the Singapore experiment. $\gamma_{\text{BB}}$ and $\gamma_r$ refer to black-body radiation induced decay of the Rydberg-state [106] and the decay time of the bare Rydberg state, respectively.

| | $\tau_{\text{BB}}/\beta^2$ | $\tau_r/\beta^2$ | $\tau_V$ | $\tau_\Delta^Z$ | $\tau_\kappa$ | $\tau_\Delta^B$ | $\tau_{g_c}$ | $\tau_{\omega_0}$ |
|---|---|---|---|---|---|---|---|---|
| $\mu s$ | 50880-361808 | 2200-15630 | 552-10472 | 15-36 | 10 | 6-20 | 10-20 | 3-10 |

Table 3: Hierarchy of timescales for all involved processes calculated from table 2.

increase the strength of interaction. This can be achieved by reducing the laser detuning to the bare Rydberg level. However, this will lead to higher inherited loss rates for the admixed state. Additionally, it could be possible to prepare an initial many-body state such that it is close to a state with an even/odd symmetry breaking. A scheme to prepare such states in extended Rydberg ensembles is in Ref. [107]. As indicated in Table 3, radiative losses set the longest timescale of the system. However, blackbody radiation induced losses can limit the coherence time in Rydberg-dressing schemes [5,106,108] as a single decay event can lead to avalanche-like losses of atoms from the trapping lattice. However, it was also pointed out [5], that such impurity Rydberg atoms could be eliminated in future experiments by using a laser quench before atom-loss occurs.

Specifically, $\gamma_{\text{BB}}$ and $\gamma_r$ refer to black-body radiation induced decay of the Rydberg-state [106] and the decay time of the bare Rydberg state, respectively. Both rates are modified by a dressing factor denoted by $\beta^2$ that determines the strength of admixing the Rydberg-level to the ground-state [5]. In both tables, $\beta^2 \in (0.012 - 0.0017)$. While spontaneous emission occurs predominantly to other ground or low-lying excited states, blackbody radiation transfers population from a virtual Rydberg excitation mostly to neighbouring high-lying states, $n \to n \pm 1$, where then a true Rydberg atom is created with a rate $\beta^2\gamma_{\text{BB}}$. By assuming stochastically triggered, instantaneous loss of all atoms in the state $|\uparrow\rangle$ good agreement with experimental data was obtained with a model to estimate the mean number $N(t)$ of remaining atoms after a dressing laser had been applied for a time $t$:

$$N(t) \approx N(0)P(t), \quad P(t) \approx \exp\left(-\frac{N(0)}{4}\gamma_{\text{BB}}\beta^2 t\right), \tag{82}$$

where $P(t)$ is the probability that no atom induced a blackbody-radiation induced loss process. An experimental determination yielded a value of $\gamma_{\text{BB}}/2\pi = 1.6 kHz$ which was almost half the literature value for $31P_{1/2}$ states [106].

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
