# Peer review of "Quantum-optical magnets with competing short- and long-range interactions: Rydberg-dressed spin lattice in an optical cavity"

_SciPost Physics, doi:SciPost Phys. 1, 004 (2016)_

## Round 1 · Referee Report · Anonymous · 2016-8-22

Strengths

1. Novel class of model, exploring competing short and long range interactions in non-equilibrium systems
2. Clear proposed signatures for experiment.

Weaknesses

1. Analysis restricted to mean field theory, with only brief discussion of beyond mean-field effects.
2. Restricted reference to other related literature on lattice open quantum systems.
3. Restricted analysis of stability to wavevectors incommensurate with periodicity of order.

Report

This paper presents a new model, considering Rydberg dressed atoms in an optical lattice, with additional coupling between hyperfine ground states (i.e. not Rydberg states) by a cavity assisted Raman process. This model combines short range interactions between the state of atoms from the Rydberg term with a long range interaction mediated by the cavity. Thus, there is a short range Ising-like interaction, as well as a transverse field proportional to the cavity light field. The dissipation terms considered cause relaxation of the atoms along the Ising axis (thus breaking the symmetry between the two spin directions in the Ising interaction). The transverse field coupling to the cavity light includes counter-rotating terms, as expected for a cavity assisted Raman processes, and this balances the photon loss terms. The authors consider a mean field theory that allows for the lattice transition symmetry to be broken by doubling the unit cell. As a result, there is a phase diagram corresponding to whether either or both of the cavity and lattice Z2 symmetries are broken.

The analysis appears correct and complete, within the scope of a mean-field decoupling considered. The work certainly adds a new an interesting direction to the study of open quantum system phase transitions using cold atoms in cavities. As the authors note in their conclusions, this paper serves as a "base case" for arrays of self-interacting atomic qubits coupled to single mode cavities. This aim is indeed reasonably satisfied, however the results in the paper leave open many questions, at least some of which I feel really require answering to fully fulfill the "outstanding-quality" requirement of scipost. These questions are as follows:

* The authors restrict to order with a period two. This may at first seem reasonable, however there are several examples of driven dissipative lattice systems where order at incommensurate wavevectors were seen [e.g. Lee et al, Phys. Rev. Lett. 110, 257204 (2013), Zou et al, Phys. Rev. Lett. 113, 023603 (2014), Schiro et al Phys. Rev. Lett. 116 143603 (2016)].

To see whether this occurs, one needs either consider longer repeat units (e.g. four or six or eight sites), or to consider linear stability analysis allowing for incommensurate wavevectors. The latter allows one to verify stability at all wavevectors directly. An approach applicable to this is described in the section "Bogoliubov theory" in Phys. Rev. A 90, 063821.

* A remarkable difference between short range and long range interactions is the role of dimensionality. i.e., for a one dimensional lattice, one can expect the order mediated by short ranged interactions to be destroyed by fluctuations, while the order mediated by the cavity should presumably survive.

This leads to interesting questions: how does the phase diagram change in such cases: does the suppression of the short range order by fluctuations lead to an enhanced region where the long range order survives? When they co-exist, is the surviving long range order able to enhance the short range order? What is the fate of the limit cycle phases when short range order is washed out?

While answering all these questions is probably beyond the scope of this paper, the discussion of this fascinating set of questions in the manuscript is as it stands rather limited.

* The citations to related work are perhaps not as complete as they might be: there is excellent coverage of relevant work on cold atoms in optical cavities, and on Rydberg states, however since the effective model ends up being a driven dissipative model on a lattice, there are many works studying very similar models (without long range coupling). As well as those noted above in connection to incommensurate phases, there are also works that have discussed phenomena such as limit cycle phases [e.g. Chan et al Phys. Rev. A 91, 051601(R)], and the breakdown of mean field theory in low dimensions [Hoenig et al, Phys. Rev. A 87, 023401 (2013), Jin et al Phys. Rev. X 6, 031011 (2016)] and bistability in Rydberg lattice systems [Lee et al Phys. Rev. Lett. 108, 023602 (2012), Hu et al, Phys. Rev. A 88, 053627 (2013)]. Establishing the connections between this work and this other active field would seem appropriate.

Requested changes

1. Discuss whether any incommensurate ordering can be ruled out, or provide stability analysis demonstrating this.

2. Discuss further expected behaviour beyond mean field theory.

3. Discuss relation to other literature on open quantum systems in lattices.

  • validity: top
  • significance: good
  • originality: high
  • clarity: high
  • formatting: good
  • grammar: excellent

Author:  Jan Gelhausen  on 2016-10-13  [id 66]

(in reply to Report 1 on 2016-08-22)

We thank the referee for the detailed report and for his positive assessment of our manuscript by highlighting that it treats a novel class of models with both short and long-range interactions in a non-equilibrium system with clear signatures for possible experiments.

We also agree with the referee concerning the criticism that she/he has raised and have added appropriate changes to the manuscript to address his/her requested changes.

We believe that with these changes made, our manuscript has benefitted in terms of clarity and quality.

Sincerely, Jan Gelhausen, Michael Buchhold, Achim Rosch and Philipp Strack

Detailed response to the referee report

Requested Change 1: Discuss whether any incommensurate ordering can be ruled out, or provide stability analysis demonstrating this.

Our response: We have performed the requested analysis and can rule out incommensurate orderings for our model at the mean-field level. In contrast to other driven-dissipative lattice models where incommensurate ordering occurs, in our model there is only the spin-spin interaction in the z-direction that depends continuously on the momentum since the atom-light interaction is sensitive to the zero-momentum components only. This means, that there are no competing, momentum-dependent interactions in contrast for example to a dissipative, anisotropic Heisenberg Hamiltonian [A1]. This is also in contrast to driven-dissipative Bose-Hubbard-type lattice models where multimode photon fields are considered that have a finite momentum-dependence [A2].

Changes to manuscript: We have added a validity analysis of the two-sublattice ansatz as an additional appendix by performing a linear stability analysis with incommensurate wave vectors as well and find no incommensurate orderings.

System Message: WARNING/2 (<string>, line 18)

Block quote ends without a blank line; unexpected unindent.

This analysis is given in a new Appendix C: Validity analysis of the even-odd sublattice Ansatz.

Requested Change 2: “Discuss relation to other literature on open quantum systems in lattices. (...) As well as those noted above in connection to incommensurate phases, there are also works that have discussed phenomena such as limit cycle phases (...), and the breakdown of mean field theory in low dimensions (...) and bistability in Rydberg lattice systems.“

Our response: We have included all of the referees suggested references and additional references at various points in the manuscript. A multitude of driven-dissipative lattice models that feature limit cycles, bistabilities and incommensurate orderings are now cited such that the model is appropriately placed within its research context.

Changes to manuscript: We have added the references [A2-A23] to our manuscript. Most of them are cited in the introduction. We have included a brief discussion about how the cited manuscripts relate to our own work and have pointed out their relevance for research about steady-state properties of out-of-equilibrium systems.

Requested Change 3: "Discuss further expected behaviour beyond mean field theory. (...) While answering all these questions is probably beyond the scope of this paper, the discussion of this fascinating set of questions in the manuscript is as it stands rather limited.“

Our response: We agree with the referee that the interplay of driving, dissipation and fluctuations in the long-time limit raises very interesting questions, especially in low-dimensional systems d <= 2.

Changes to manuscript: In the new section Sec IV D we have provided a qualitative discussion on how selected features of the phase diagram might change in lower dimensionality d=1 (not the case of our present paper which considers d=2 arrays). In d=2, we believe the main features to be robust. We also reference further technical possibilities to capture quantum fluctuations in large, driven-dissipative interacting quantum spin systems; all of these are under development at the time of writing.

References [A1] Lee et al, "Unconventional magnetism via optical pumping of interacting spin systems“, Phys. Rev. Lett. 110, 257204 (2013) [A2] Zou et al, “Implementation of the dicke lattice model in hybrid quantum system arrays,” Phys. Rev. Lett. 113, 023603 (2014) [A3] Chan et al, “Limit-cycle phase in driven-dissipative spin systems,” Phys. Rev. A 91, 051601 (2015) [A4] Wilson et al,“Collective phases of strongly interacting cavity photons,” arXiv:1601.06857 (2016). [A5] Keeling et al “Collective dynamics of bose-einstein condensates in optical cavities,” Phys. Rev. Lett. 105, 043001 (2010) [A6] Piazza et al, “Self-ordered limit cycles, chaos, and phase slippage with a superfluid inside an optical resonator,” Phys. Rev. Lett. 115, 163601 (2015) [A7] Rossini et al, “Steady-state phase diagram of a driven qed-cavity array with cross-kerr nonlinearities,” Phys. Rev. A 90, 023827 (2014) [A8] Boité et al, “Bose-hubbard model: Relation between driven-dissipative steady states and equilibrium quantum phases,” Phys. Rev. A 90, 063821 (2014) [A9] Boité et al, “Steady-state phases and tunneling-induced instabilities in the driven dissipative bose-hubbard model,” Phys. Rev. Lett. 110, 233601 (2013) [A10] Zou et al, “Implementation of the dicke lattice model in hybrid quantum system arrays,” Phys. Rev. Lett. 113, 023603 (2014) [A11] Mendoza-Arenas et al, “Beyond mean-field bistability in driven-dissipative lattices: Bunching-antibunching transition and quantum simulation,” Phys. Rev. A 93, 023821 (2016) [A12] Tomadin et al, “Signatures of the superfluid-insulator phase transition in laser-driven dissipative nonlinear cavityarrays,” Phys. Rev. A 81, 061801 (2010) [A13] Tomadin et al, “Nonequilibrium phase diagram of a driven and dissipative many-body system,” Phys. Rev. A 83, 013611 (2011) [A14] Everest et al, “Atomic loss and gain as a resource for nonequilibrium phase transitions in optical lattices,” Phys. Rev. A 93, 023409 (2016). [A15] Nissen et al, “Nonequilibrium dynamics of coupled qubit-cavity arrays,” Phys. Rev. Lett. 108, 233603 (2012) [A16] Weimer, “Variational principle for steady states of dissipative quantum many-body systems,” Phys.Rev. Lett. 114, 040402 (2015). [A17] Weimer, “Variational analysis of driven-dissipative rydberg gases,” Phys. Rev. A 91, 063401 (2015) [A18] Jin et al, “Cluster mean-field approach to the steady-state phase diagram of dissipative spin systems,” Phys. Rev. X 6, 031011(2016) [A19] Strogatz, Nonlinear Dynamics And Chaos: With Applications To Physics, Biology, Chemistry, And Engineering (Studies in Nonlinearity), 1st ed., Studies in nonlinearity (Perseus Books Group). [A20]Schirmer et al, “Stabilizing open quantum systems by markovian reservoir engineering,” Phys. Rev. A 81, 062306 (2010). [A21] Maghrebi et al, “Nonequilibrium many-body steady states via keldysh formalism,” Phys. Rev. B 93, 014307 (2016). [A21] Hu et al, “Spatial correlations of one-dimensional driven-dissipative systems of rydberg atoms,” Phys. Rev. A 88, 053627 (2013). [A23] Schiró et al, “Exotic attractors of the nonequilibrium rabi-hubbard model,” Phys. Rev. Lett. 116, 143603 (2016)

---

## Round 1 · Referee Report · Anonymous · 2016-9-11

Strengths

- The manuscript deals with interesting competition between local and long-range interactions between atoms in a cavity
- The form of the model is novel
- The physics treated is strongly motivated by new possibilities in experiments

Weaknesses

- The authors do not discuss the dynamical timescales accessible in their suggested implementation
- The authors do not discuss spontaneous emissions in dressed Rydberg levels

Report

The authors treat a system of atoms in a cavity, which interact with their neighbouring atoms directly, and are also coupled via the cavity mode. Considering the presence of dissipation both via the cavity mode and via spontaneous decay, they determine the mean-field phase diagram for steady states, and discuss the interplay between magnetic ordering and super-radiance in the cavity mode. Different combinations of phases are also found to have signatures in the spectrum of light emitted through the cavity mode. They investigate the dynamics in each regime in detail, and describe how this setup could be engineered with multi-level atoms in a cavity, with direct interactions due to Rydberg dressing of one of the internal states.

This paper is very timely given the significant recent progress in experiments with atoms in cavities, and in working with interactions between atoms in Rydberg states. I found the model studied here, with the corresponding mean-field solutions, to be an interesting prototypical starting point for investigating the interplay between long-range interactions mediated by a cavity and direct short range spin-spin interactions. The paper is also generally very accessible, and the calculations well explained. Overall, I feel that the paper is a high-quality manuscript, although there are a few areas in which improvements could be made to clarify aspects of the potential implementation, and what might be expected beyond mean-field calculations.

Specifically, I have the following comments:

1) The proposed implementation in Section II is interesting, and the authors identify very specific atomic states. Given that, I would find it very helpful if the authors could provide more information on the corresponding estimated timescales and frequency scales for the Hamiltonian and Liouvillian, as well as the dynamical timescales. Especially, there are a number of competing timescales involved, and it is not clear to me whether or not all of the phases are necessarily accessible in the setup the authors describe.

2) A potential issue with Rydberg dressing implementations is that corresponding spontaneous emissions can never be completely avoided, and then there can be a significant decay cascade, in which the atoms spend some time in relatively long-lived Rydberg levels before returning to the ground state. I would find it very useful, in combination with the estimates of realistic frequency and timescales, if the authors could estimate the corresponding timescales for such potentially problematic decays. It is possible that these could dominate spontaneous emission processes of the form that the authors include in their model in more detail. Naturally, including the full atomic physics of the Rydberg levels is not realistic in this manuscript, and the analysis of the model is useful anyway. But I feel it would strengthen the manuscript significantly if the authors indicated this potential issue, and estimated how the timescales on which such imperfections were important compared with other timescales in the system.

3) The authors calculate a lot of results in their mean-field approximation. I think it would be helpful if they could comment on potential corrections beyond this, and in which limits they expect this to work well, and where they expect dynamics beyond mean-field to play an important role.

As a minor point, while the manuscript is mostly very well written, there are a number of small typos and grammatical errors that should be corrected, e.g.:
- "master equations equations"
- "such as for example"
- "competing interactions potentials"
- "not to loose excitations"
- Commas in unusual places ("equations for both, the...")
- References to Fig. 7 early in the document, where probably Fig. 1 is intended (or Fig. 7 should come earlier in the manuscript)

Requested changes

- The authors should estimate the dynamical timescales of their proposed implementation, and identify the accessible regions of the phase diagram
- The authors should comment on the validity of the mean-field solutions and potential effects beyond these
- Minor grammatical erros should be corrected

  • validity: top
  • significance: high
  • originality: high
  • clarity: high
  • formatting: excellent
  • grammar: excellent

Author:  Jan Gelhausen  on 2016-10-13  [id 67]

(in reply to Report 2 on 2016-09-11)

We thank the referee for the detailed and considered report.

We will address the requested changes below and we are thankful to the referee for pointing out to us areas where our manuscript can benefit from additional information.

We believe that with these changes made, our manuscript has benefitted in terms of clarity and quality.

Sincerely, Jan Gelhausen, Michael Buchhold, Achim Rosch and Philipp Strack

Detailed response to the referee report

Requested Change: Estimate the time and frequency scales for the Hamiltonian and Liouvillian dynamics in order to determine whether or not all mean-field phases are accessible

Our response: We have performed this comparison of time scales using data from two recently performed experiments. One was performed on a 2d Ising Hamiltonian with an interaction between Rydberg-dressed ground states carried out by Zeiher et al. [B1], where typical time and frequency scales were measured in detail.

System Message: WARNING/2 (<string>, line 15)

Definition list ends without a blank line; unexpected unindent.

For the estimation of typical optical and spin-light frequency scales, we rely upon the experiment on the Dicke superradiance transition of a gas of ultracold atoms coupled to a high-finesse optical cavity by Baden et al. [B2]. As our model combines both quantum optical implementation schemes, we consider this a reasonable starting point. From this procedure, we conclude that the spin-spin interaction of Rydberg-dressed ground state atoms currently is still small compared to the optical frequency scales such as the cavity boosted atom-light coupling. We hope that this can be remedied when the detuning to the bare Rydberg level is reduced. However this might not be without consequence for the inherited decay rates of the Rydberg-dressed ground-state.

Changes to manuscript: We have put together a hierarchical list containing typical values for all parameters of our model and have added an appropriate discussion in a new appendix D: Hierarchy of time and energy scales and problematic Rydberg decays

Requested Change: Discuss spontaneous emission in dressed Rydberg levels with a view to decay cascades.

Our response: We have discussed these Rydberg decay effects in the revised version: Decay-cascades in Rydberg dressing schemes can indeed limit the time for coherent interactions [B3]. While spontaneous emission of dressed ground-states is predominantly directed towards other ground-states in the hyperfine-structure manifold, blackbody induced radiation can create a true Rydberg excitation by moving population from a dressed state to a high-lying Rydberg state. This has severe consequences as the resulting dipole-dipole interactions can lead to avalanche like losses of atoms that are in the Rydberg-dressed state as also observed in the experiment by Zeiher et al. Here, a single atom-event has consequences for an entire spin lattice. However, Zeiher et al. have expressed hopes to prevent cascade like losses by employing stroboscopic dressing or by laser quenching impurity Rydberg atoms.

Changes to manuscript: The table of timescales also involves a timescale for blackbody induced radiative decay. We have also added a discussion in appendix D about problematic black-body induced radiative decay in Rydberg dressing schemes. We have added References [B2,B3,B4] to the manuscript.

Requested Change: Discuss in which limits the mean-field theory of this model is expected to work well and where dynamics beyond mean-field can be expected.

Our response: As we were asked to address this point in-depth before, we kindly refer to our response given to the review posted on 2016-08-22. Changes to manuscript: See our response to the review posted on 2016-08-22.

Requested Change: As a minor point, while the manuscript is mostly very well written, there are a number of small typos and grammatical errors that should be corrected. Our response: We thank the referee for pointing out to us various typos. Changes to the manuscript: We have corrected the typos. References [B1]: J. Zeiher et al “Many-body interferometry of a rydberg-dressed spin lattice,” Nature Physics 38, 35 (2016). [B2]: M. P. Baden et al, “Realization of the dicke model using cavity-assisted raman transitions,” Phys. Rev. Lett. 113, 020408 (2014). [B3]: E.A.Goldschmidt et al, “Anomalous broadening in driven dissipative rydberg systems,” Phys. Rev. Lett. 116, 113001 (2016). [B4]: T. Pohl et al, “Dynamical crystallization in the dipole blockade of ultracold atoms,” Phys. Rev. Lett. 104, 043002 (2010).

---

## Round 1 · Referee Report · Tony Lee · 2016-9-15

Strengths

1. The model is novel, and the competition between long- and short-range interactions is interesting.
2. The analysis is thorough.
3. The results could be relevant to experiments.

Weaknesses

1. Doesn't mention validity of mean field.
2. Some of the language and figures are confusing.

Report

The manuscript presents a new model in which a short-range spin model interacts with a dissipative cavity model, thereby leading to a competition between short-range and long-range interactions. The authors do a mean-field analysis and find a new phase that includes antiferromagnetic spin correlations and a superradiant cavity mode. They discuss how the various phases can be detected via the cavity spectrum.

The study seems to be the first to consider the combination of short and long-range interactions. The results are interesting and the authors make a convincing case for how the model could be realized in an experiment, so the paper will probably stimulate many theoretical and experimental studies into this subject. Thus, I believe the paper is suitable for publication in SciPost after the below changes are made.

Requested changes

There should be some discussion about when the mean-field approximation is valid, especially the dependence on dimensionality.

Eq. 2: Explain why one shouldn't just take the rotating-wave approximation

Pg. 5: The sentences after "The two states captured..." need to be rephrased.

Sec IB: The word "coexistence" may confuse readers because it suggests that two distinct phases are bistable, whereas the AFM+SR is really one phase. I suggest avoiding the word "coexistence" throughout the paper (unless it's really bistable).

Sec IB: Why does the ramp speed of g matter? Doesn't the system just go to the steady state?

Sec IC: The text refers to Fig 9, but should this be Fig 3?

Eq. 7: Define v

Fig 4: The shading for phases 2,4 don't exactly match the legend

Fig 5: The shading for phase 6 doesn't exactly match the legend

Fig 6a: The lines should have shorter dashes

Eq. 23-26: The authors appear to be using the Keldysh formalism. They should provide a reference. Also, the retarded Green function and its inverse should be written out explicitly, since they are used later on.

Eq. 37,38,40: The denominators should be written as absolute values.

Sec IVB2: This section is a bit confusing, because the solutions are denoted (v1,v2), but there can be four or six poles. The language here should be clarified.

Fig 9: The legends should have a bounding box, or else the lines appear to disappear (especially in (a)). Also, in (a) and (b), the red lines appear to have only 3 peaks, while they are supposed to have 6 and 4 peaks. I assume the other peaks aren't visible due to the range of the y axis. Unless the authors can find a way to make all the peaks visible, I suggest removing the red lines, or else it's confusing why there's only 3 peaks.

  • validity: top
  • significance: top
  • originality: high
  • clarity: good
  • formatting: excellent
  • grammar: excellent

Author:  Jan Gelhausen  on 2016-10-13  [id 65]

(in reply to Report 3 by Tony Lee on 2016-09-15)

Dear Tony Lee,

We thank you for the detailed report and positive assessment of our manuscript by highlighting that our results for the novel model of a short-range spin model coupled to a lossy cavity are interesting and potentially relevant to experiments.

We also agree with the referee in his view that our analysis could trigger further theoretical and experimental investigation in this new type of models that should be within experimental reach.

We have addressed the requested changes and have adjusted the relevant parts accordingly. We believe that with these changes made, our manuscript has benefitted in terms of clarity and quality.

Sincerely, Jan Gelhausen, Michael Buchhold, Achim Rosch and Philipp Strack

Detailed response to the referee report

Requested Change: Eq. 2: Explain why one shouldn't just take the rotating wave approximation

Our response: With the help of cavity-assisted Raman transitions, the atom-light coupling of the two-level atom can be boosted to be on the same order of magnitude as for instance the cavity detuning. A treatment in time-dependent perturbation theory is therefore not adequate. Co- and counter rotating terms are naturally induced by the Raman scheme. While the energy-conserving terms describe coherent exchange of spin and light excitations, the counter-rotating terms describe the scattering of photons from the pump-laser into the cavity and vice versa. The latter is needed to guarantee that the lossy cavity can maintain a fixed number of excitations in the steady-state. Changes to manuscript: We have added the above clarification below Eq.2 where the atom-light Hamiltonian is presented.

Requested Change: Eq. 23-26: The authors appear to be using the Keldysh formalism. They should provide a reference. Also, the retarded Green function and its inverse should be written out explicitly, since they are used later on.

Our response: An explicit expression for the inverse of the retarded Green function is given in Eq. 26. Since this allows the calculation of the polariton poles, we believe this to be sufficient. Changes to manuscript: We have added an appropriate reference for the Keldysh framework, see [C1].

Requested Change: Sec IVB2: This section is a bit confusing, because the solutions are denoted (v1,v2), but there can be four or six poles. The language here should be clarified.

Our response: We believe that after our revision, there should not arise any confusion concerning the labelling or number of poles. Changes to manuscript: We have rewritten the section IVB2 to improve on its clarity.

Requested Change: Sec IB: Why does the ramp speed of g matter? Doesn't the system just go to the steady state?

Our response: We have added a discussion about estimated time and frequency scales for the Hamiltonian and the Liouvillian in order to estimate what regime of the phase diagram could experimentally be accessible. Changes to manuscript: The changes are summarised in the new appendix D: Hierarchy of energy scales and problematic Rydberg decays.

Requested Change: Sec IB: The word "coexistence" may confuse readers because it suggests that two distinct phases are bistable, whereas the AFM+SR is really one phase. I suggest avoiding the word "coexistence" throughout the paper (unless it's really bistable).

Our response: The AFM+SR phase is really one phase. Changes to the manuscript: We have revised the manuscript and avoid using the word coexistence in order to avoid confusion.

Requested Change: Minor amendments for Figures 4,5,6,9 and Eqs 2,37,38,40

Our response: We thank the referee for pointing out to us various typos. Changes to the manuscript: We have made all suggested changes to improve clarity.

Requested Change: There should be some discussion about when the mean-field approximation is valid, especially the dependence on dimensionality

Our response: As a similar question was asked in the report posted on 2016-08-22, we kindly refer to our response in the aforementioned reply to avoid repeating the information. Changes to the manuscript: We refer to the report posted on 2016-08-22.

References [C1] Kamenev, Alex "Field Theory of Non-Equilibrium Systems“, 1st edition, Cambridge University Press (2011)

---

## Round 2 · Author Response

We have addressed all points mentioned by the referees and have supplied a detailed response to every referee report. We are thankful for the referees suggestions and we believe that with these changes made our manuscript has benefitted in terms of clarity.

---

## Round 2 · List of Changes

added appendix C (Validity analysis of the even-odd sub lattice Ansatz), added appendix D (Hierarchy of energy scales and problematic Rydberg decays), added Sec IVD (Discussion of beyond mean-field effects), added references to various driven-dissipative lattice models

---

## Editorial Decision

published